# Sources of dehydration fluids underneath the Kamchatka arc

Yunchao Shu [1,2,3] ✉, Sune G. Nielsen [1,4] ✉, Veronique Le Roux [4], Gerhard Wörner [5], Jerzy Blusztajn[1,4] & Maureen Auro[1]

Fluids mediate the transport of subducted slab material and play a crucial role in the generation of arc magmas. However, the source of subduction-derived fluids remains debated. The Kamchatka arc is an ideal subduction zone to identify the source of fluids because the arc magmas are comparably mafic, their source appears to be essentially free of subducted sediment-derived components, and subducted Hawaii-Emperor Seamount Chain (HESC) is thought to contribute a substantial fluid flux to the Kamchatka magmas. Here we show that Tl isotope ratios are unique tracers of HESC contribution to Kamchatka arc magma sources. In conjunction with trace element ratios and literature data, we trace the progressive dehydration and melting of subducted HESC across the Kamchatka arc. In succession, serpentine (<100 km depth), lawsonite (100–250 km depth) and phengite (>250 km depth) break down and produce fluids that contribute to arc magmatism at the Eastern Volcanic Front (EVF), Central Kamchatka Depression (CKD), and Sredinny Ridge (SR), respectively. However, given the Tl-poor nature of serpentine and lawsonite fluids, simultaneous melting of subducted HESC is required to explain the HESC-like Tl isotope signatures observed in EVF and CKD lavas. In the absence of eclogitic crust melting processes in this region of the Kamchatka arc, we propose that progressive dehydration and melting of a HESC-dominated mélange offers the most compelling interpretation of the combined isotope and trace element data.

Mass transfer between the surface of the Earth and the deep interior takes place at subduction zones. In particular, subduction has been linked to the release of fluid-rich materials from subducted slabs into the mantle wedge, leading to the generation of hydrous magmas sampled by arc volcanoes[1,2]. It has long been held that the mobilization of $H_2O$-rich sediments and hydrothermally altered oceanic crust (AOC) during subduction modifies the composition of the mantle wedge and affects the geochemical signatures of arc magmas[3–9]. Compared to mid-ocean ridge basalts (MORBs), the most noticeable features of arc magmas include high $H_2O$ contents, high abundances of fluid-mobile elements (e.g. Ba, Sr, Pb, B), and characteristic elemental fractionations (e.g. high Ba/Th and Sr/Nd, low Ce/Pb)[3–7]. However, the mechanisms that control the transfer of fluid components into the overlying mantle wedge remain debated.

Some models posit that the subducted slab remains sufficiently intact during subduction processes so that elements are mobilized from sediments and AOC separately, releasing melts and fluids, respectively, once critical mineral dehydration and melting reactions

[1]NIRVANA (Non-traditional Isotope Research for Various Advanced Novel Applications) Laboratories, Woods Hole Oceanographic Institution, Woods Hole, MA 02543, USA. [2]The Pheasant Memorial Laboratory for Geochemistry and Cosmochemistry, Institute for Planetary Materials, Okayama University, Misasa, Tottori 682-0193, Japan. [3]CAS Key Laboratory of Crust-Mantle Materials and Environments, School of Earth and Space Sciences, University of Science and Technology of China, Hefei 230026, China. [4]Department of Geology and Geophysics, Woods Hole Oceanographic Institution, Woods Hole, MA 02543, USA. [5]Geowissenschaftliches Zentrum der Universität Göttingen, Göttingen 37073, Germany. ✉e-mail: yshu@ustc.edu.cn; snielsen@whoi.edu

are crossed along the slab-top[4–6,10–12]. Depending on the age and temperature of the down-going lithosphere, fluids may primarily come from dehydration reactions of AOC[5,6,12], or derive from underlying serpentinite and re-equilibrated with minimally altered lower oceanic crust[13]. Another alternative is that fluids derive from dehydration of mélange rocks[14], i.e. a mechanical, potentially buoyant, the mixture of different slab lithologies[15–17]. Field evidence based on subducted lithologies that have been tectonically exhumed shows that the slab-mantle interface may consist of several kilometer-thick mélange zones of physically mixed sediments, AOC, and serpentinized mantle[16,18–20]. Melting and dehydration of mélange rocks has been hypothesized as an alternative mechanism for the generation of arc magmas, which can account for both radiogenic isotope and trace element ratios observed in arc lavas[14,16,17,21,22].

Depending on the source of the fluids, and the timing of their release, geochemical predictions can be made as to whether fluids are primarily derived from AOC[5,6,12], serpentinites and minimally altered lower oceanic crust[13], or mélange rocks[14]. Disentangling these components is simplified in arcs where the recycled sediment component is minimal, and where the oceanic crust component can be tracked with certainty. In this respect, samples from an across-arc transect in the Kamchatka arc (Fig. 1) are ideal because of the high abundance of mafic intermediate- to high-K arc basalts and Mg-andesites, and because there is no evidence for significant sediment input into their mantle source regions[23–28]. The northern part of the Kamchatka arc has formed directly above the subducted Hawaii-Emperor Seamount Chain (HESC), which consists of plume-related sub-marine basalts erupted on the ca. 87 Ma old oceanic plate[29]. It has been argued that the fluid input from the subducted HESC dominates the fluid component of Kamchatka arc magmas in this region and, compared to other arc settings, is the cause for the high magma production rate in particular in the Central Kamchatka Depression[23,24,26].

The Kamchatka arc has been extensively studied previously[23–27,30–37] and we here only provide a broad overview as context for the studied samples. Mafic lavas were selected primarily in portions of the Kamchatka arc where the HESC is subducting, from Holocene volcanoes across the arc front (Eastern Volcanic Front: EVF), the Central Kamchatka Depression (CKD) and the Sredinny Ridge (SR) in the back arc. The slab depth varies from 100–140 km below the EVF to 100–200 km below the CKD and 300–400 km below the SR[38]. The CKD represents an intra-arc rift structure where the highest magma production in any arc setting around the world is observed[24]. Only samples from the volcano Bakening (Southern section) are outside the core zone of HESC subduction. The other volcanoes studied are Gamchen, Kizimen, Komarov and Schmidt (all EVF), Nikolka, Tolbachik and Kluchevskoy (all CKD), and Achtang, Esso and Ichinsky (all SR), and fall inside of the broad zone of subduction of the HESC. All selected Kamchatka arc lava samples have previously been investigated for their major and trace (including chalcophile) elements (Supplementary Table 1), and a subset of samples had also previously been measured for O-, Li-, Sr-, Nd-, Hf-, Zn-, and Pb-isotope ratios (Supplementary Table 2)[23,24,31,33,34]. The selected lavas are, except for three magmatically more evolved samples, either basalts or basaltic andesites with MgO > 3 wt.% and SiO$_2$ ranging from 47.4 to 56.0 wt.% (Supplementary Table 1).

To constrain the composition of potential slab components, we also analysed sediment cores and HESC-basalts recovered from

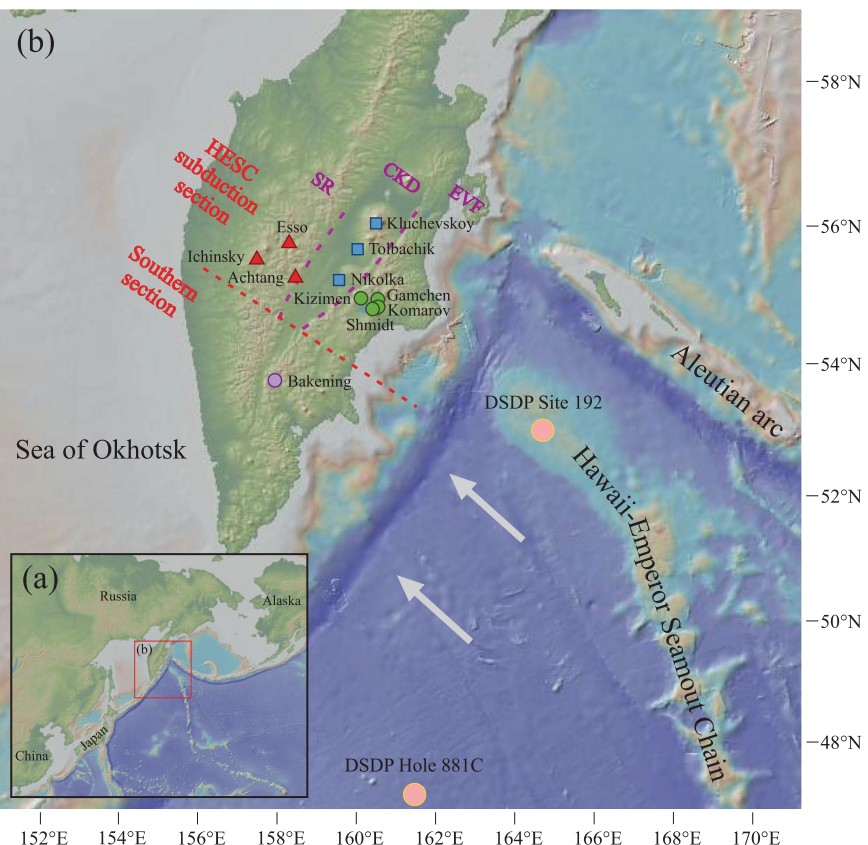

**Fig. 1 | Location of Kamchatka subduction zone. a** Location of Kamchatka arc and surrounding areas. **b** Map of the Kamchatka arc and Hawaii-Emperor Seamount Chain (HESC). Solid circle, square and triangle indicate the sampled volcanoes from Eastern Volcanic Front (EVF), Central Kamchatka Depression (CKD) and Sredinny Ridge (SR), respectively. Also shown is the location of two drill cores from which samples were analyzed in this study to constrain the sediment and HESC input flux to the arc. Approximate plate motion direction is represented by arrows[24]. Perpendicular to the plate motion direction, Kamchatka arc was divided into two sections: HESC subduction zone (EVF, CKD and SR); south of HESC subduction zone (Bakening volcano indicated by purple circle). The image of landform made with GeoMapApp 3.6.10 (www.geomapapp.org) / CC BY (https://creativecommons.org/licenses/by/4.0/) / CC BY[123].

previous DSDP and ODP drill cores. DSDP Site 192 was drilled on top of the Meiji Seamount that is part of the HESC and most proximal to Kamchatka (Fig. 1). The core covers a total sediment thickness of 1044 m from seafloor to the basaltic basement and consists of diatomaceous silty clay, diatom ooze/clay, (calcareous) claystone and chalk[39]. The samples selected for the study represent the major lithological units identified in the entire sediment column (Supplementary Table 3). Pyroxene-plagioclase diabasic basalts, which represent the underlying HESC oceanic crust at DSDP site 192, were recovered from 1044 to 1057 m below sea floor[39]. Five samples covering the entire length of recovered core were selected for this study (Supplementary Table 3). ODP Hole 881 C was drilled in the abyssal plains to the south of the HESC. The core shows a total sediment thickness of 364 m and comprises mostly clayey diatom ooze[40]. The water depth at ODP Hole 881 C is 5530 m[40], while it is 2980 m at the Meiji seamount (part of the HESC)[41]. Sediment thickness on the Meiji seamount is about 1044 m[39], which would indicate a minimal thickness of HESC basalts of roughly 1500 m. The underlying oceanic crust was not drilled at ODP Hole 881 C. The 10 selected samples represent the two major units described in the entire sediment section[40].

In order to investigate sources of fluids in the Kamchatka arc, we present new thallium (Tl) isotope compositions for 64 Holocene arc lavas from 11 different volcanoes from a SE-NW transect between 55–56° latitude across Kamchatka where the Hawaii-Emperor Seamount Chain is being subducted. Unlike radiogenic isotopes, Tl stable isotopes do not vary over time and do not require time-dependent corrections and evolution models. Furthermore, although Tl is a fluid-mobile element, Tl isotopes do not fractionate during dehydration of AOC, partial melting and fractional crystallization[42–44], which enables their use as direct tracers of subducted components that may carry distinct Tl isotope signatures[45–48]. As sediments contain significantly higher Tl concentrations than bulk oceanic crust, Tl isotope variations in arcs are usually controlled by sediment inputs[45–47]. In the northern part of the Kamchatka arc, however, the distinct composition of the subducted HESC can be particularly well tracked by Tl isotopes throughout the subduction cycle, aided by negligible sediment input in that region. We also report new elemental and isotope data (Tl, Sr, Nd, Pb) for 5 HESC samples and 15 sediment samples collected from the Deep-Sea Drilling Project (DSDP) Expedition 19 Site 192, and for 10 sediment samples collected from Ocean Drilling Program (ODP) Expedition 145 Hole 881 C (see Fig. 1 for location). These two sites represent offshore analogues for subducted components under the Kamchatka arc.

In this study, our results reveal systematic Tl isotope and trace element ratio variations from arc front to the back-arc that are exclusively consistent with gradual mass loss from the top of the slab whereby an HESC-rich mélange containing residual lawsonite below the arc front and residual phengite in the Central Kamchatka Depression control isotope and trace element systematics in these two regions. Finally, an HESC-free oceanic crust in which phengite breaks down is the primary slab material found in the back-arc region.

## Results and discussion
### Elemental and radiogenic isotope signatures of sediments and HESC
Elemental (major and trace elements) and radiogenic isotope systematics (Sr, Nd, Pb) are reported in Supplementary Table 4 and 5, respectively. The chemical compositions of sediments and HESC samples are consistent with their lithological descriptions (Supplementary Table 3). For sediments, diatom-rich samples have high $SiO_2$ content, chalk-rich samples have high $CaO$ content, and clay-rich samples have high $Al_2O_3$ content. Hydrothermally altered HESC samples exhibit low Ba/Rb and Nb/U, and high Li/Dy, consistent with the addition of Rb, U, and Li from seawater during hydrothermal fluid circulation at low temperatures (<150 °C)[49,50].

The radiogenic isotope compositions of the sediments were only measured at Site 192 and these are broadly similar to values previously found in ODP Holes 881C[51,52]. The DSDP 192 value of $^{87}Sr/^{86}Sr$ ~ 0.707 (Supplementary Table 5) is less radiogenic than the value of $^{87}Sr/^{86}Sr$ ~ 0.711 that was previously inferred for Site 881[53], based on data from DSDP Sites 579 and 581 located significantly to the south of Kamchatka[54].

The Sr and Nd radiogenic isotope compositions of the HESC samples resemble those previously reported for Meiji Seamount samples[55] and other seamounts along the HESC in the Pacific[56]. However, $^{206}Pb/^{204}Pb$ and $^{207}Pb/^{204}Pb$ ratios are more variable towards more radiogenic values (Supplementary Table 5). These radiogenic values are consistent with large enrichments in U due to hydrothermal alteration, which have led to significant radiogenic ingrowth since the emplacement of the Meiji Seamount ~85Ma[55].

### Thallium systematics in sediments and HESC
Thallium has two isotopes, $^{203}Tl$ and $^{205}Tl$. Its isotopic composition is reported in epsilon units relative to the NIST-SRM 997 Tl reference material in parts per 10,000 as:

$$\varepsilon^{205}Tl = 10,000 \times \left( {^{205}Tl}/{^{203}Tl}_{sample} - {^{205}Tl}/{^{203}Tl}_{SRM997} \right) / \left( {^{205}Tl}/{^{203}Tl}_{SRM997} \right)$$
(1)

Thallium isotope variations in sediments and HESC are reported in Supplementary Table 5 and are shown in Fig. 2. The sediments from DSDP Site 192 (i.e. on top of HESC seamounts) display $\varepsilon^{205}Tl$ ranging from −4.7 to +3.4, and elemental Tl concentrations ranging from 57 to 329 ng/g. Pelagic sediments from ODP Hole 881 C range from $\varepsilon^{205}Tl$ = −2.5 to +0.9, and from 72 to 337 ng/g Tl (Fig. 2). The weighted average Tl isotope compositions of Site 192 and Hole 881 C sediments are $\varepsilon^{205}Tl$ = −2.0 and −1.2, respectively (Supplementary Table 6), and were calculated using the concentration and isotope data obtained for individual samples combined with the thicknesses and densities of the different sediment units present in all the drill cores[39,40]. These values are indistinguishable from that of the mantle ($\varepsilon^{205}Tl$ = −2.0 ± 0.5[57–59]) and continental crust ($\varepsilon^{205}Tl$ ~ −2[60]), and likely reflect the dominance of terrigenous input in both locations. Minor amounts of manganese oxide may be responsible for heavier Tl isotope values in some of the samples, as marine authigenic Mn oxides are strongly enriched in thallium (with up to ~ 100 μg/g) and record heavy Tl isotope compositions ($\varepsilon^{205}Tl$ = +12.8 ± 1.2[61–63]). However, the only three sediment samples with appreciably heavier Tl isotope compositions also record some of the lowest Tl concentrations (Fig. 2), most likely due to a significant carbonate component in these sediments (Supplementary Table 4) that is essentially devoid of Tl[62], which render these sediments relatively unimportant in the overall Tl budget of the sediment column.

In contrast, altered HESC basalts display consistently heavy Tl isotope compositions, from $\varepsilon^{205}Tl$ of +0.9 to +5.0, and concentrations ranging from 12 to 61 ng/g (Fig. 2). These heavy Tl isotope values are unlikely to reflect direct precipitation of Mn oxides on the basalts as the basalts were carefully handpicked under binocular microscope from internal fragments of every sample to avoid any surface contamination. In addition, Tl shows no positive co-variation with other trace metals that are usually hosted in Mn oxides such as Co, Ni, and Mo (Supplementary Fig. 1). Instead, the heavy Tl isotope compositions likely indicate significant contribution from pelagic sediments with marine authigenic Mn oxides in the mantle source region of the HESC basalts as has previously been suggested for Hawaiian lavas based on Hf and Tl isotopes[58,64,65]. It should be noted that the HESC samples record, on average, heavier Tl isotope compositions than recent Hawaiian lavas[58,65]. Thus, it is possible this reflects a temporal evolution of the Hawaii-Emperor plume or potentially minor Mn oxide deposition on some of our HESC samples (Supplementary Fig. 1), although we

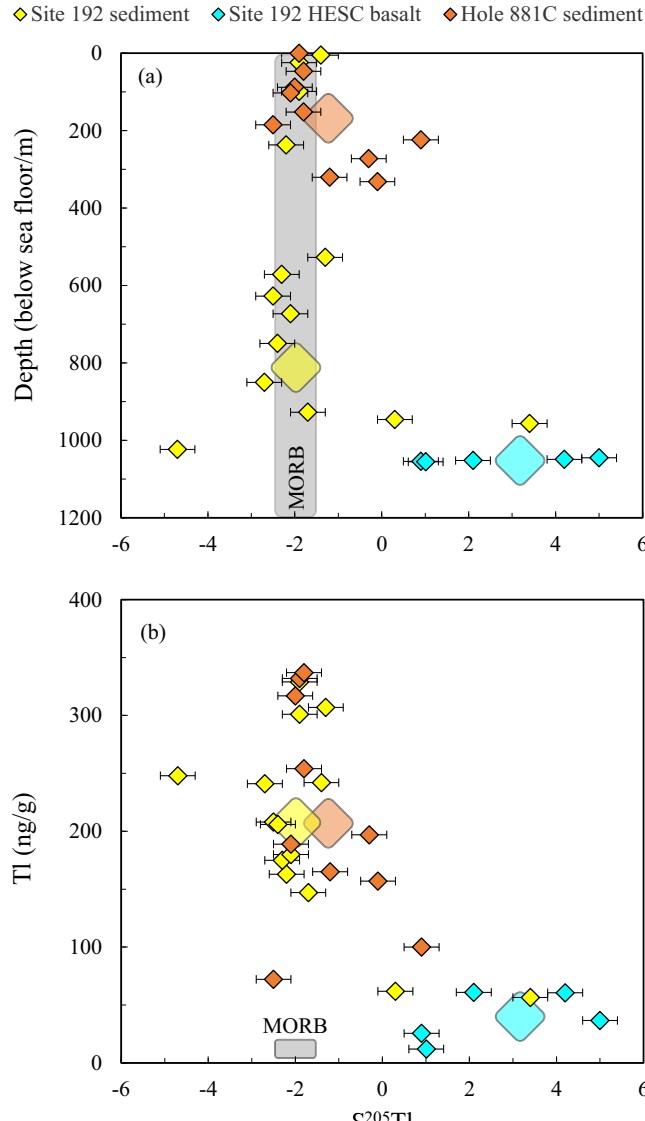

**Fig. 2 | Thallium concentration and isotope characteristics of subducting materials outboard of Kamchatka arc.** Thallium isotope compositions plotted against (**a**) depth and (**b**) Tl concentrations for sediments from Deep Sea Drilling Project (DSDP) Site 192 and Ocean Drilling Program (ODP) Hole 881 C and Hawaii-Emperor Seamount Chain (HESC) basalts from DSDP Site 192. The lighter shaded areas indicate the average Tl concentration and isotope compositions of mid-ocean ridge basalt (MORB)[58,59,66]. Large symbols indicate the concentration weighted average Tl isotope compositions and concentrations for each of the two sediment cores and HESC.

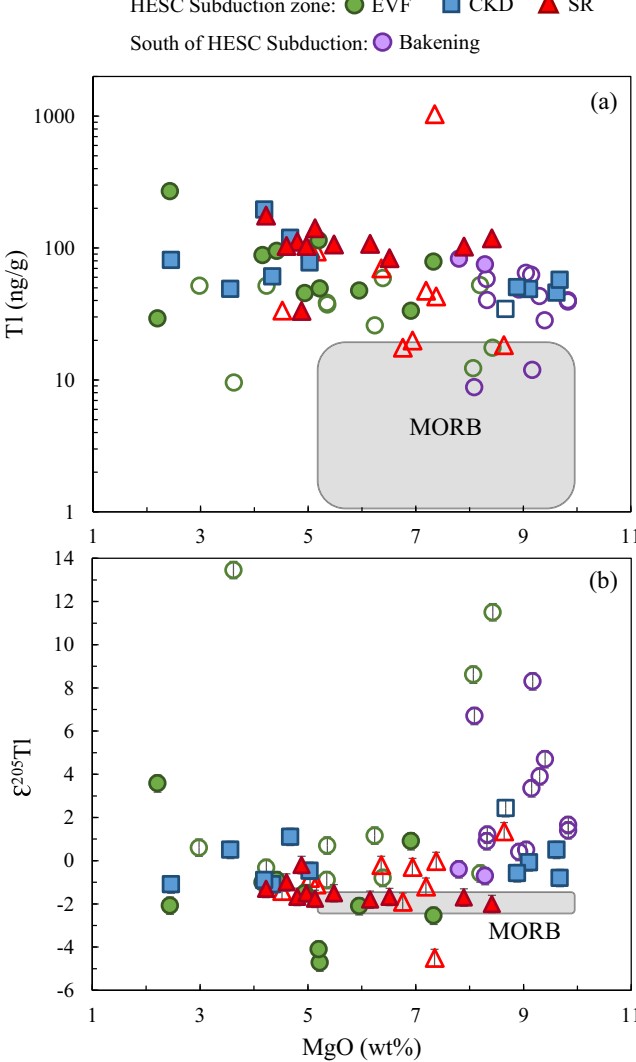

**Fig. 3 | Thallium concentration and isotope characteristics of Kamchatka arc lavas.** MgO (wt%) plotted against (**a**) Tl concentrations and (**b**) Tl isotope compositions for all Kamchatka lavas investigated in this study. Also shown is the mid-ocean ridge basalt (MORB)[58,66,67,75]. Note that the legend colors follow that of the map in Fig. 1 and are the same for all subsequent figures with lavas from multiple locations. Open symbols represent filtered Kamchatka lavas for degassing and ocean-island-basalt (OIB) components in their mantle source.

consider the latter less likely given the lack of positive correlation between Tl and other indicators of Mn oxide deposition (Supplementary Fig. 1). The unique Tl isotope characteristic of HESC basalts will be used to trace this component in the source of Kamchatka lavas and its relationship to other tracers of fluid release.

**Thallium systematics in Kamchatka arc lavas**
Thallium concentrations in the Kamchatka arc lavas range from ~9 to 1040 ng/g (Supplementary Table 1), which is significantly higher and more variable than in MORB (1 to 20 ng/g[66,67]). The Kamchatka arc lavas display a large range of Tl isotope compositions and the heaviest Tl isotope values in arc lavas documented to date, with $\varepsilon^{205}Tl = -4.7$ to +13.5 (Supplementary Table 2). The vast majority of lavas exhibit Tl isotope values comparable to global arcs[45–47,68], ranging from $\varepsilon^{205}Tl \sim -3$

to +3 (Fig. 3), ranging to heavier values than the depleted MORB mantle ($\varepsilon^{205}Tl \sim -2$[57–59]) and offshore sediments analyzed in this study (weighted average Tl isotope compositions of $\varepsilon^{205}Tl = -2.0$ to $-1.2$). However, the Kamchatka arc lava data broadly overlap with the heavy Tl isotope compositions measured in the HESC samples ($\varepsilon^{205}Tl = +0.9$ to +5.0).

**Deciphering degassing and source effects in Kamchatka lavas**
Thallium concentrations and isotope compositions of arc lavas are potentially affected by several processes that can obscure the primary source signature obtained from mixing between the mantle wedge and slab components. These are assimilation and fractional crystallization (AFC), magma degassing, and post-eruptive alteration[45–47,68]. Thallium abundances in the Kamchatka arc lavas do not correlate with MgO (Fig. 3). Broad correlations between Tl abundances and MgO content have been observed in other volcanic arcs, e.g. the Ryukyu, Central America and Tonga-Kermadec arcs[45,47], but there the effects of crystal fractionation are mainly detected in highly fractionated samples (MgO

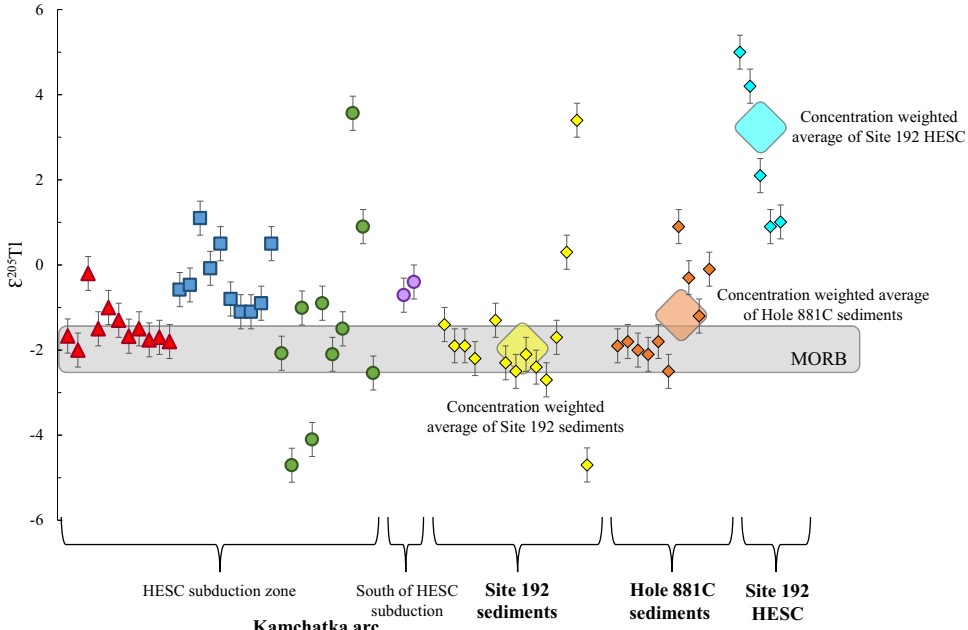

**Fig. 4 | Thallium isotope compositions of Kamchatka lavas that have been filtered for degassing and deep ocean-island-basalt (OIB) components (see discussion for details), sediments and Hawaii-Emperor Seamount Chain (HESC) basalts from the two drill cores.** Grey area in chart denotes the Tl isotope composition of the mid-ocean ridge basalt (MORB)[58,59]. Large symbols indicate the concentration weighted average Tl isotope compositions for each of the two sediment cores and HESC.

< 3 wt.%), which have been avoided here for the purposes of this study. The Kamchatka arc lavas also exhibit Th/Rb and Ce/Tl similar to unaltered lavas from other island arcs (Supplementary Information). Of syn- or posteruption processes, neither AFC nor post-eruptive alteration have played a significant role (Supplementary Information), whereas degassing has recently been shown to affect Tl isotope signatures in some Kamchatka lavas[69]. Therefore, such samples need to be identified and excluded here. We use the observed correlation of a Tl degassing trend towards high Ce/Tl and heavy (positive) Tl isotope signatures and the experimentally determined kinetic Tl isotope and Ce/Tl fractionation during degassing[69] to identify the least degassed lavas from each Kamchatka volcano (Supplementary Fig. 2). According to this test, several samples from the volcanoes Bakening, Gamchen, Komarov, Kluchevskoy, Shmidt and Esso suffered significant Tl degassing and their Tl isotope values are reported in italics in Supplementary Table 2. The remaining undegassed samples range from $\varepsilon^{205}Tl$ of −4.7 to +3.6 (Fig. 4) and we only consider these in the subsequent discussion and figures.

It is notable that samples from Ichinsky volcano exhibit some of the highest Ce/Tl, yet do not appear to have experienced significant degassing given their light Tl-isotope ratios (Supplementary Fig. 2j). However, based on relatively high Nb/La, Churikova et al. (2001) suggested that some magmas at Ichinsky and other back-arc volcanoes are sourced from a mantle wedge that was enriched by an OIB-type mantle component[24]. Such a mantle signature would be expected to be associated with variably high Ce/Tl[48,58,59], and elevated Dy/Yb due to the presence of garnet[27,32,70]. Indeed, a significant fraction of the Ichinsky lavas display elevated Ce/Tl (on average 1331) and Dy/Yb >2 (Supplementary Fig. 3) consistent with the presence of a deep, garnet-rich component in their source[27,32,48,58,59,70]. Therefore, we disregard samples that likely reflect an OIB-like component, and only rely on samples that have similar Ce/Tl and Dy/Yb to the rest of the Kamchatka arc series.

### Contributions of sediments and HESC-AOC in the Kamchatka arc
Previous studies have concluded that the mantle source of Kamchatka lavas across the HESC subducted zone is not significantly affected by subducted sediments, whereas fluids from the HESC- and AOC-basalts play a more important role[23–28]. New high-precision Pb isotope data for sediments and HESC basalts allow for a quantitative assessment of the relative importance of these two components in the source (Fig. 5). The strongly radiogenic $^{206}Pb/^{204}Pb$ values for HESC basalts are consistent with U addition during hydrothermal alteration as is also observed in other AOC sections[71], which implies that any HESC component should be characterized by radiogenic $^{206}Pb/^{204}Pb$. Previous studies inferred that the HESC component had substantially lower $^{206}Pb/^{204}Pb$ than what we observe here and that fluids released from HESC and/or AOC that mixed with Pacific MORB mantle could account for all Pb isotope variations in the Kamchatka arc[28]. However, Pb isotope data for HESC basalts, sediments and Pacific MORB mantle reveal that these cannot be combined to form the Pb isotope array of Kamchatka arc lavas (Fig. 5). Instead, the combination of new and previously published Pb isotope data imply that Indian MORB mantle is the dominant component in the Kamchatka mantle wedge. This is consistent with previous studies[33], and similar to what has been observed for other western Pacific subduction zones (e.g. Marianas, Ryukyu, Izu-Bonin, Kurile)[45,72–74]. The full range of Pb isotope variations in the Indian MORB field is very large[75], but given that many Kamchatka arc lavas exhibit relatively unradiogenic Pb isotope compositions (Fig. 5), the mantle wedge cannot realistically overlap with the more radiogenic portions of the Indian MORB field[75]. Furthermore, peridotite xenoliths derived from the Avachinsky volcano in the southern part of Kamchatka peninsula are consistent with a homogenous, highly refractory mantle wedge[76], which would be consistent with unradiogenic Pb isotope compositions. Although some mantle xenoliths from Kharchinsky and Shiveluch volcanoes (both located in the northern part of the CKD) display Pb isotopic ratios overlapping with the Pacific MORB field, these samples also contained extremely high Pb concentrations, which was interpreted to reflect infiltration of the mantle wedge beneath Kamchatka by Pb-rich slab-derived fluids[77]. Therefore, they are likely not representative of the mantle wedge itself[77], which would explain the mismatch with arc lavas (Fig. 5). In addition, some literature Kamchatka arc lavas have $^{206}Pb/^{204}Pb$ ~ 18.1–18.2, which is

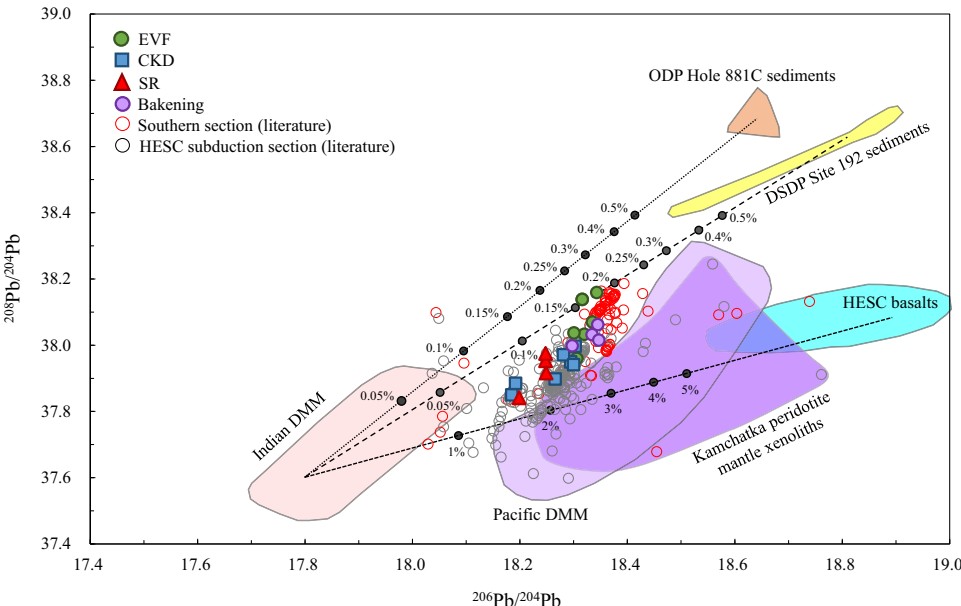

**Fig. 5 | Lead isotope characteristics for Kamchatka lavas and subducting materials outboard of Kamchatka arc.** $^{206}Pb/^{204}Pb$ versus $^{208}Pb/^{204}Pb$ diagram for Kamchatka lavas that have been filtered for deep ocean-island-basalt (OIB) components from Ichinsky volcano lavas. Also shown are the Pacific and Indian depleted MORB mantle (DMM)[75,124], Kamchatka peridotite mantle xenoliths infiltrated by fluids[77], sediments and Hawaii-Emperor Seamount Chain (HESC) from Deep Sea Drilling Project (DSDP) Site 192 and Ocean Drilling Program (ODP) Hole 881 C and mixing lines between Indian DMM and bulk sediments from ODP Hole 881 C and DSDP Site 192, respectively, as well as HESC basalts from DSDP Site 192. Numbers in mixing lines represent bulk sediment or HESC additions by weight. Literature data for Kamchatka arc are from the GEOROC database[125].

lower than both the mantle xenolith and Pacific MORB field (Fig. 5). Therefore, the mantle wedge beneath Kamchatka must have primary Pb isotope characteristics of $^{206}Pb/^{204}Pb < 18.1$ and $^{208}Pb/^{204}Pb < 37.6$, which is only observed for the Indian MORB field (Fig. 5).

The Pb isotope mass balance calculations demonstrate that bulk sediments indeed are only a minor component in Kamchatka arc lavas (<0.2% by weight), as has been suggested earlier, whereas bulk HESC contributions can be accounted for by 1–5% by weight, depending on the assumed Pb isotope composition of the refractory mantle wedge component (Fig. 5). It can also be seen that magmas forming where the HESC subducts contain an even smaller sediment component than the Southern Kamchatka lavas (Fig. 5). This observation is consistent with a thicker oceanic crust and less efficient sediment subduction associated with the HESC[23,24,27]. These calculations assume bulk components, so that if sediment and HESC contributions occurred in the form of melts or fluids, the slab components required to explain the Pb isotope variations would be even smaller because Pb is preferentially mobilized into slab-derived melts and fluids[7,11]. Previous calculations suggested that more than 90% of Pb in the Kamchatka samples could be attributed to fluids derived from dehydration of subducted basalts[24]. However, quantifying the exact contributions of each slab component in Pb isotope space would require more robust Pb partitioning data, and is beyond the scope of this study. Importantly, Pb isotopes cannot distinguish between a mélange model where components first mix, then melt and dehydrate, and a model where slab components melt and/or dehydrate separately, and are subsequently added to the mantle wedge as metasomatic agent(s). Below, new Tl isotope data are used in combination with trace element ratios to distinguish these two models of slab material transport.

### Subduction of Hawaii-Emperor Seamount Chain recorded by Tl isotopes
The Kamchatka arc lavas that have been filtered for degassing and back-arc OIB components (Supplementary Table 7), display Tl isotope compositions ($\epsilon^{205}Tl$ from −4.7 to +3.6) ranging from typical MORB values to heavier (with the exception of two samples that are lighter).

Given that sediments outboard of Kamchatka trench display much higher Tl concentrations than HESC basalts (Fig. 2b), significant sediment inputs could in principle affect Tl isotope variation in the Kamchatka arc, as observed in other arcs[45–47]. However, mixing between marine sediments ($\epsilon^{205}Tl = -2$ to −1.2) and the sub-arc mantle wedge ($\epsilon^{205}Tl = -2$) cannot account for the range of Tl isotope variations in Kamchatka arc lavas (Fig. 4), which is also consistent with the Pb isotope data (Fig. 5) and previous evidence for a sediment-poor source of Kamchatka arc lavas[23–28]. Similarly, the Indian MORB mantle wedge is also incapable of accounting for the heavy Tl isotope compositions as it has previously been inferred to be characterized by $\epsilon^{205}Tl = -2$, based on Tl isotope data from the Mariana and Ryukyu arcs[45,68]. Instead, HESC ($\epsilon^{205}Tl = +0.9$ to +5.0) is likely the primary cause for the heavy ($\epsilon^{205}Tl > -1$) Tl isotope composition. The lighter Tl isotope compositions observed in two lavas ($\epsilon^{205}Tl = -4.1$ and −4.7) could represent "normal" AOC underlying the HESC, as AOC usually ranges from $\epsilon^{205}Tl$ of −10 to −2[57]. It is also notable that the volcano Bakening, which sits to the south of the axis of HESC subduction underneath Kamchatka, exhibits Tl isotope compositions within error of pelagic sediments outboard of the Kamchatka arc (Fig. 4). Consistent with slightly higher sediment contributions in the southern part of the Kamchatka arc based on Pb isotopes (Fig. 5), these Tl isotope data hint at a minor influence of sediments in the Bakening volcano. However, the overall small Tl isotope variations between sediments and the mantle wedge render it speculative to further constrain any significant influence of sediment subduction based on Tl isotopes.

### Fluids from serpentinite and HESC-AOC dehydration under the Eastern Volcanic Front
Previous studies have shown that EVF arc lavas display high $\delta^{11}B$[26,78], high B/La[30], and enrichment in chalcophile elements such as As, Sb, and Pb[31]. Although enrichments in chalcophile elements may be linked to the breakdown of sulfides, EVF lavas are also enriched in Sr and Ba that are not present in sulfides. Alternatively, sediments and AOC also contain high abundances of B, chalcophile elements, Sr, and Ba, and thus it has been suggested that fluids expelled from these components

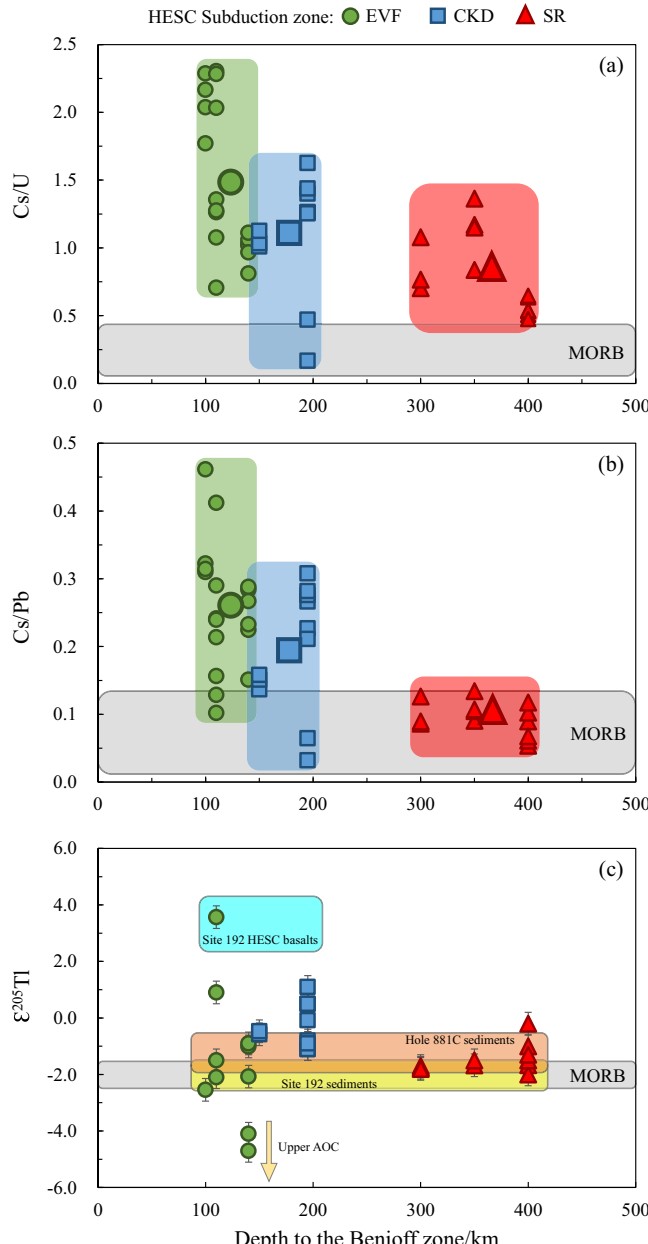

**Fig. 6 | Variation of trace element ratios and thallium isotope compositions in Kamchatka arc lavas within the zone of Hawaii-Emperor Seamount Chain (HESC) subduction along slab depth.** a Cs/U, (**b**) Cs/Pb and (**c**) $\varepsilon^{205}$Tl values versus depth to the Benioff zone. The depth to the Benioff zone is from ref. [38]. The Kamchatka lavas in (**a**) and (**b**) have been filtered for deep ocean-island-basalt (OIB) components for Ichinsky volcano. The Tl isotope compositions of Kamchatka lavas in (**c**) have been filtered for degassing and deep OIB components (see discussion for details). Also shown are the fields of fresh mid-ocean ridge basalt (MORB)[57,58,67], Deep Sea Drilling Project (DSDP) Site 192 sediments (yellow bar), Ocean Drilling Program (ODP) Hole 881 C sediments (orange bar) and HESC basalts. The Tl isotope compositions of the sediment and basalt components were calculated as concentration weighted average of the individual sediments and basalts analyzed (Supplementary Table 6). Regular low-temperature altered oceanic crust (AOC) displays $\varepsilon^{205}$Tl values of −2 to −15[57] and is indicated by an arrow.

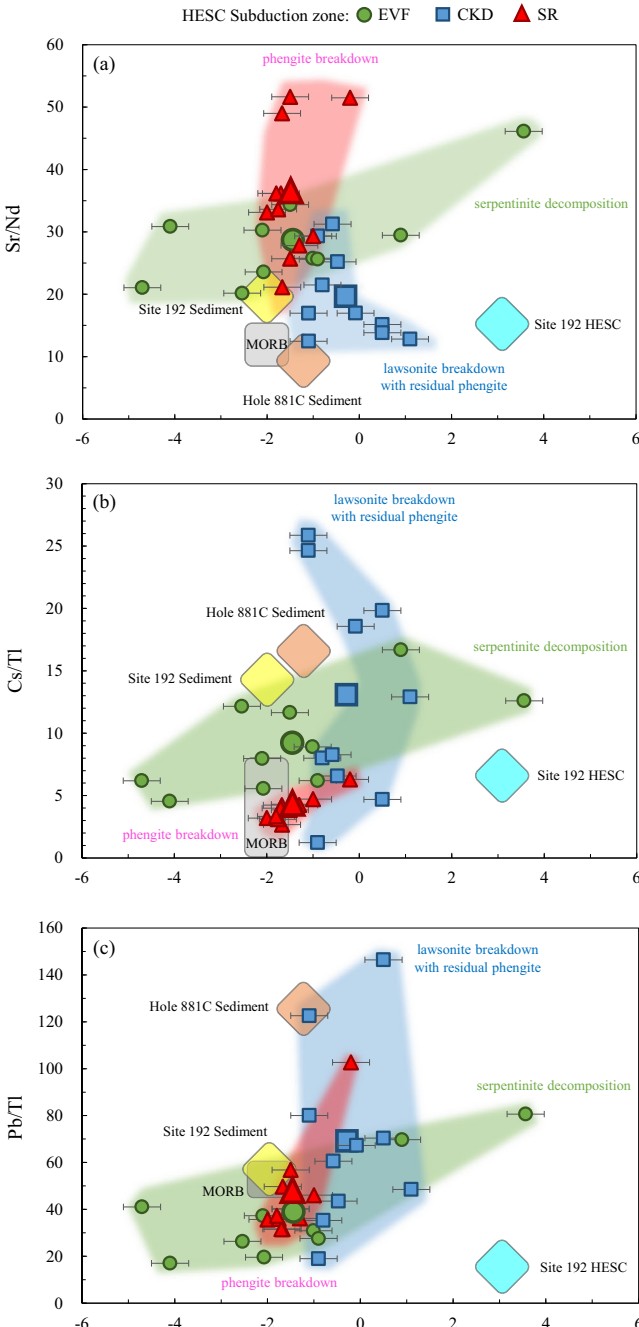

**Fig. 7 | Trace element and Tl isotope compositions indicating progressive breakdown of different minerals at sub-arc depth.** Thallium isotope compositions plotted versus (**a**) Sr/Nd, (**b**) Cs/Tl and (**c**) Pb/Tl for Kamchatka lavas within the zone of Hawaii-Emperor Seamount Chain (HESC) subduction. The lavas have been filtered for degassing and deep ocean-island-basalt (OIB) components (see discussion for details). Green, blue and red areas in each plots indicate serpentinite decomposition, lawsonite breakdown and phengite breakdown beneath Eastern Volcanic Front (EVF), Central Kamchatka Depression (CKD) and Sredinny Ridge (SR), respectively. See discussion for details. Large symbols indicate the average trace element ratio and Tl isotope compositions for each of the three sample groups.

could hydrate the overlying mantle to form a serpentinite layer enriched in these elements[79]. In the case of Kamchatka, sediment contributions are minor. Therefore, HESC and underlying AOC would be the main contributors to fluids that feed potential serpentinite formation in the forearc mantle, as hinted by both heavy (HESC) and light (AOC) $\varepsilon^{205}$Tl values in the EVF lavas (Figs. 6 and 7).

The EVF lavas also display high Cs/U and Cs/Pb (Fig. 6), pointing to the presence of the mineral lawsonite in the HESC-AOC portion of the slab. Indeed, Pb, Cs and U are all fluid-mobile elements[80], but Pb and U are preferentially retained in lawsonite[81–83], implying that high Cs/U and Cs/Pb are produced when fluids are released from a lawsonite-

bearing slab as long as sediment components, that are often characterized by high Cs/U and Cs/Pb[53,84], are absent. In addition, phengite is the primary mineral that can accommodate Tl in the subducted slab at arc magma generation depths[42,44,85]. Phengite contains K as an essential constituent (appr. 10 wt% $K_2O$)[86–88]. Although sediments enriched in K do not play an important role in the Kamchatka arc magma generation[23–28] and fresh oceanic basalts contains very low $K_2O$[67,89], HESC material measured here (Supplementary Table 4) and previously[55], have K contents that are comparable to the overlying sediments. The high K contents of HESC material are likely due to their origin from an enriched mantle plume as well as a result of hydrothermal alteration at low temperature[49,90–93]. Thus, phengite is likely to be stable in slab components that contain HESC basalts. Our previous studies have also shown that Tl concentration and isotope data for samples of subducted basalts are inconsistent with any systematic Tl isotope fractionation during partial removal of Tl from a phengite-bearing assemblage[42,44]. However, aqueous fluids released in the presence of phengite from the HESC-AOC would contain little Tl[44,85]. Therefore, Tl cannot be transported from HESC to an overlying serpentinite layer exclusively by fluids and is likely transported in a HESC-derived melt. However, melting of HESC basalts is not likely at the pressure and temperature conditions at the slab-mantle interface below the EVF[7,94].

An alternative model that is consistent with all the above data involves dehydration and melting of a mélange that consists of HESC, underlying AOC, and overlying serpentinites. Mélange melting has been documented to occur at lower temperatures than what is required for eclogitized oceanic crust[21,22], although mélange melting could also occur in diapirs rising from the slab due to decompression (Fig. 8). This model best accounts for variations in Tl isotopes in EVF

lavas because mélange melts may mobilize sufficient Tl, given that fluids in this portion of the arc are likely Tl-poor[44,85]. The partitioning of Tl during mélange melting is presently unknown, but given that phengite is likely a minor phase in a mélange, it is possible that bulk melt-residue partition coefficients are substantially higher than the low fluid-residue partition coefficients found for high-pressure metamorphic rocks[44]. To avoid the uncertainties relating to absolute Tl partitioning, we rely on the relative Tl content of the lavas as monitored by Cs/Tl. The chemical behavior of Tl is considered to group with the heavy alkali metals K, Rb, and Cs, due to their similar ionic radii and charge[95–97]. However due to Cs being a larger cation than Tl[96], Cs fits more poorly in the phengite structure than Tl, making it less compatible in phengite[42,47]. On average, EVF lavas display higher Cs/Tl than SR lavas consistent with the presence of residual phengite.

The high $\delta^{11}B$ and enrichments in B-Sr-Ba-chalcophile elements in EVF lavas are also consistent with this model as serpentine within the mélange would breakdown at depths of 90–110 km[98–101], which is similar to the slab depth beneath the EVF volcanoes (Fig. 6).

## Fluids from lawsonite breakdown under the Central Kamchatka Depression

As discussed above, the heavy Tl isotopes observed in EVF Kamchatka arc lavas reflect a contribution from HESC material that is also visible in the Tolbachik and Nikolka volcanoes of the CKD. These volcanoes are centered right on the path of the projected subducted HESC[24]. Heavy Tl isotopes in CKD are associated with on average lower Sr/Nd, Cs/U and Cs/Pb compared to EVF, and higher Pb/Tl (Figs. 6 and 7). Lawsonite is a main host for light rare earth elements (e.g. Nd), Sr, U, Th, and Pb[81–83], but does not contain significant quantities of alkali metals. Therefore, lawsonite breakdown should be associated with high Pb/Tl

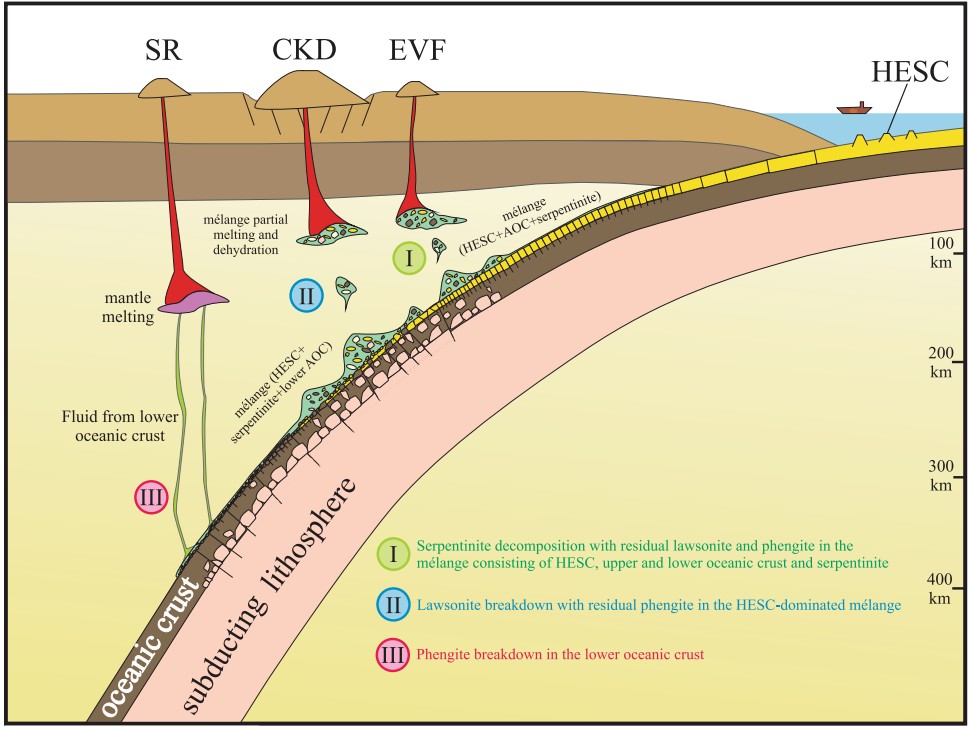

**Fig. 8 | Cartoon of the mélange model of slab material transport in Kamchatka subduction zone.** (I) Below Eastern Volcanic Front (EVF), the mélange contains Hawaii-Emperor Seamount Chain (HESC), upper and lower oceanic crust and serpentinite. Decomposition of serpentine with residual lawsonite and phengite in the mélange account for the highest fluid-mobile chalcophile trace element concentrations and δ[11]B values as well as Cs/U, Cs/Pb and B/Nb, together with relatively high Cs/Tl and U/K and low Sr/Nd and Ba/Th. (II) Underlying Central Kamchatka

Depression (CKD), the mélange consists of HESC, serpentinite and/or lower oceanic crust. Breakdown of hydrous lawsonite and residue of phengite in the HESC of the mélange results in relatively low Ba/Th and Sr/Nd and high Cs/Tl, U/K and Pb/Tl in CKD lavas. (III) Beneath Sredinny Ridge (SR), the HESC-rich mélange layer was substantially mobilized before fluids contributing to SR are released. Breakdown of phengite in the lower oceanic crust results in high Ba/Th and Sr/Nd and low Cs/Tl, U/K and Pb/Tl in SR lavas. See discussion for details.

and relatively low Cs/Pb, Cs/U and Sr/Nd[81]. As suggested in previous studies, lawsonite must break down below the CKD, consistent with other elemental and isotopic tracers[31], whereas phengite likely remains stable in the source of fluids that fed the magma source of CKD volcanoes at greater depths.

The lawsonite-out reaction should dominate water release from a subducted slab surface depth of 100 km and up to 250 km[78,102] and this is consistent with observed slab depth below the CKD volcanoes (Fig. 6). However, lawsonite-derived fluids contain very little Tl (~1 ng/g)[44]. Therefore, the Tl isotope compositions of lawsonite-derived fluids likely re-equilibrate during their ascent through the mantle wedge and obtain mantle-like values ($\epsilon^{205}$Tl = −2). On the other hand, dehydration and melting of HESC-dominated mélange can account for the HESC-like Tl isotope signatures.

In addition, it has been argued that serpentinite dehydration controls high $\delta^{11}$B values in some CKD volcanoes[78], although the enrichment is not as clear as for the EVF lavas. It has also been argued that HESC dehydration explains heavy $\delta^{18}$O and $\delta^{7}$Li signatures in some CKD lavas[23,31]. However, serpentinites contain negligible quantities of Tl and would not affect the Tl budget or Tl isotope signatures of the fluids[103], demonstrating that lawsonite and phengite must play a dominant role for Tl underneath the CKD region. Lawsonite and phengite are not present in serpentinites, but have been reported in mélange rocks[16,104,105]. The simultaneous tracking of lawsonite, phengite ± serpentinite in the source region of CKD lavas is, therefore, consistent with the presence of a phengite-lawsonite bearing mélange dominated by HESC material.

### Fluids from phengite breakdown under the Sredinny Range

As opposed to the CKD lavas, mantle-like Tl isotope compositions ($\epsilon^{205}$Tl = −2) in the Sredinny Ridge back-arc portion of the arc are associated with high Sr/Nd, and low Cs/Tl and Pb/Tl (Fig. 7). These elemental fractionations are not compatible with lawsonite breakdown, suggesting that lawsonite has been consumed and is no longer present in the slab below these back-arc volcanoes. This geochemical inference is consistent with a slab surface depth of >300 km determined for these volcanoes (Fig. 6). However, phengite breakdown can explain the quantitative release of residual Tl and Cs from the slab, consistent with low Pb/Tl, as well as MORB-like Cs/Tl. In the absence of lawsonite, high Sr/Nd would also be expected in fluids sourced from a residue that contains REE-host minerals (e.g. allanite)[81].

These observed trace element signatures in the back-arc region are associated with mantle-like Tl isotope values. Since phengite is the main host of Tl in the subducted oceanic lithosphere, Tl-bearing fluids derived from phengite breakdown would contain relatively high Tl concentrations that would likely preserve their Tl isotopic signature on their way to the surface. This prediction is also consistent with the observation that SR lavas contain, on average, 33% higher Tl concentrations than EVF and CKD lavas (Fig. 3 and Supplementary Table 7). The fact that the fluids derived from phengite breakdown are associated with mantle-like Tl isotope values implies that the slab underneath the SR cannot be composed of HESC-dominated material. Thus, the HESC material is absent at greater depths and has likely been mechanically removed from the slab-top before reaching deeper back-arc slab depths, since slab melting is not recorded under this section of the CKD volcanoes in the Kamchatka arc[23,24,26,28]. The mantle-like Tl isotope signatures observed for SR lavas would reflect a mantle wedge in the back arc that was overprinted by a relatively small fluid contribution[24] from lower oceanic crust, as lower oceanic crust typically is associated with Tl isotope compositions similar to the unmodified mantle[106]. Although K concentrations in lower oceanic crust are typically lower than in HESC basalts[107], phengite is commonly found in a wide range of eclogites[42,108], suggesting that phengite may still play a role on controlling Tl and alkali metal mobilization.

### Sources and processes of fluid release below the Kamchatka arc

The sources of fluids across the section of the Kamchatka arc where the HESC is subducted have been identified using a combination of Tl isotope systematics and trace element compositions in EVF, CKD and SR lavas. Collectively, Tl and B isotopes, B, Cl, Pb, LILE, LREE and chalcophile elements attest to the progressive dehydration and melting of a serpentine-HESC-AOC mélange from arc front to back (Fig. 8). Underneath the EVF, breakdown of serpentine would commence, triggering partial mélange melting, while lawsonite and phengite would still be stable. The resulting high $\delta^{11}$B values[26,30,31,78] and the highest Cs/U and Cs/Pb, reflect the serpentine breakdown and residual lawsonite, respectively. These are accompanied by a large range of Tl isotope ratios ($\epsilon^{205}$Tl = −4.7 to +3.6) derived from the HESC and low-T AOC. Progressive dehydration and melting underneath the CKD occurs at moderate depth in the slab (100–250 km below the arc), and reflects lawsonite breakdown in an HESC-dominated mélange. These reactions produce magmas with heavy Tl isotopes ($\epsilon^{205}$Tl = −1.1 to +1.1), accompanied by high Cs/Tl and low Sr/Nd. Finally, dehydration and melting below the SR at deeper depths in the slab (>250 km below the arc) likely reflects phengite breakdown in the lower oceanic crust after the earlier mechanical removal of the HESC-dominated mélange below the CKD (Fig. 8). These reactions produce fluids and melts that are characterized by mantle-like Tl isotope values ($\epsilon^{205}$Tl = −2.0 to −0.2), low Cs/Tl and high Sr/Nd.

Our comprehensive approach that includes trace element ratios, stable and radiogenic isotopes, highlight the necessity to combine multiple tracers of fluid release to identify the phases that control elemental fractionations and mass transfer in arc magmas.

## Methods

### Samples

The Kamchatka arc lavas were obtained as powders and processed as received. We used aliquots of powders for Kamchatka arc lavas that were previously investigated for major and trace element and radiogenic isotope studies. Sediment samples from DSDP Site 192 and ODP Leg 145 Hole 881 C were obtained from relevant ODP core repositories and dried in an oven at 80 °C for 24 h before powdering in the Peter Hooper GeoAnalytical lab at Washington State University. HESC basalt samples were fractured into mm-size chips, and handpicked under a binocular microscope to select pieces devoid of possible contamination with sediments precipitated from seawater. These chips were ultra-sonicated in Milli-Q water for an hour to remove any dust or superficial contaminants. Powdered samples, including Kamchatka arc lavas and sediments, and HESC basalt chips, of 0.1–0.3 g were dissolved in a 5:1 mixture of concentrated distilled $HNO_3$ and HF on a hotplate for at least 24 h. They were then dried and fluxed several times using a 1:1 mixture of concentrated distilled $HNO_3$ and HCl until the fluorides, which formed in the first step, were completely dissolved. Following this, samples were dried again on a hotplate and dissolved in appropriate acid matrices. Thallium, Sr, Nd and Pb were separated from the matrices of samples in the NIRVANA (Non-traditional Isotope Research for Various Advanced Novel Applications) clean lab at Woods Hole Oceanographic Institution (WHOI) using published ion exchange chromatographic methods[109–114]. Total procedural blanks including sample dissolution and column chemistry for Tl, Sr, Nd and Pb were always <11 pg, <100 pg, <30 pg and <50 pg respectively, which is insignificant compared with the minimum amounts of each element processed.

### Thallium isotopic analysis

The Tl isotopic compositions were measured on a Thermo Finnigan Neptune multicollector inductively coupled plasma mass spectrometer (MC-ICP-MS) at the WHOI Plasma Facility. External correction for mass discrimination to NIST SRM 981 Pb and reference material

bracketing of the NIST SRM 997 Tl were applied for measurement of Tl isotopic compositions[109,110]. Thallium concentrations were determined by measuring the $^{205}$Tl/$^{208}$Pb ratios during the isotopic measurements and quantified from the known quantity of NIST SRM 981 Pb added to each sample. The secondary reference material of United States Geological Survey (USGS) reference basalt powder BHVO-1 was also processed with every set of unknowns and had isotope compositions of $\varepsilon^{205}$Tl = −3.6 ± 0.4 (2 s, n = 61), in excellent agreement with previous work ($\varepsilon^{205}$Tl = −3.5 ± 0.5[103] and $\varepsilon^{205}$Tl = −3.6 ± 0.4[42,45]). The Tl concentration of BHVO-1 was found to be 39 ± 5.7 ng/g (2 s) also in agreement with the previous studies[42,45,103]. Therefore, we use these uncertainties of ±0.4 $\varepsilon^{205}$Tl-units (2 s) and ±15% (2 s) throughout this study as our best estimates of the total external error on individual Tl isotope and concentration measurements, respectively, because these uncertainties account for all possible sources of error including sample dissolution, ion exchange chromatography and mass spectrometric procedures[109].

### Sr, Nd and Pb isotopic analysis

The Sr, Nd and Pb isotopic compositions of seamount basalts and selected sediments and arc lavas were determined using the Neptune MC-ICP-MS, located in the Plasma Facility at WHOI. For Sr isotope measurements, $^{82}$Kr, $^{83}$Kr and $^{85}$Rb were monitored to correct for the interference of $^{86}$Kr on $^{86}$Sr and $^{87}$Rb on $^{87}$Sr. All results were corrected against the Sr reference material NIST SRM 987 ($^{87}$Sr/$^{86}$Sr = 0.710240[115]). For Nd isotope measurements, $^{147}$Sm and $^{149}$Sm were monitored to correct for the isobaric interferences of $^{144}$Sm on $^{144}$Nd. The $^{143}$Nd/$^{144}$Nd values for JNdi-1 were adapted to the La Jolla $^{143}$Nd/$^{144}$Nd of 0.511847[116], using a ratio of 1.000503[117]. All Nd results in this study were corrected against JNdi-1 value of $^{143}$Nd/$^{144}$Nd = 0.512104. All Pb results were corrected using NIST SRM 981 Pb reference material ($^{206}$Pb/$^{204}$Pb = 16.9356, $^{207}$Pb/$^{204}$Pb = 15.4891, $^{208}$Pb/$^{204}$Pb = 36.7006)[118]. The well-characterized USGS reference material BHVO-1 was measured as unknown. Its results for Sr, Nd are $^{87}$Sr/$^{86}$Sr = 0.703466 and $^{143}$Nd/$^{144}$Nd = 0.51298 respectively, and after recalculating to correction based on ref. 119, Pb isotope compositions for BHVO-1 are $^{206}$Pb/$^{204}$Pb = 18.694, $^{207}$Pb/$^{204}$Pb = 15.574, $^{208}$Pb/$^{204}$Pb = 38.361, which are within error of former work[120].

### Trace and major element analysis

Trace element concentrations of DSDP sediment and seamount basalt samples were measured using an ICAP-Q ICP-MS at WHOI. Calibration curves were determined by analyzing four rock reference materials (AGV-1, BCR-1, BHVO-1 and BIR-1) with recommended concentrations[121]. The precision and accuracy of these analyses is between 5 and 10% based on the measurement of additional dissolutions of the same USGS rock reference materials as unknowns during the same analysis runs.

Major elements for the DSDP Site 192 and ODP Leg 145 Hole 881 C samples (both sediment and Seamounts) were determined by X-ray fluorescence (XRF) in the Peter Hooper GeoAnalytical lab at Washington State University using previously described techniques[122]. Precision and accuracy for these measurements are better than ~3% for the most abundant major elements (e.g. primarily Si, Mg, Fe, Ca, Al and Na) and better than ~10% for the minor elements with concentrations <1% (e.g. primarily Ti, P, Mn).

### Data availability

The authors declare that the data generated or analyzed during this study are included in this published article and its Supplementary Information files.

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

## Acknowledgements

This study was financially supported by grants from the National Natural Science Foundation of China (NSFC) (Grant No. 41903008) and Chinese Postdoctoral Science Foundation (Grant No. 2019M660153) to Y.S., NSF (Grant No. EAR-1829546) to S.G.N. and NSF (Grant No. EAR-1855302) to V.L.R. Tanja Churikova and Frank Dorendorf are thanked for their contributions during sampling in the field.

## Author contributions

Analyses were made by Y.S. and J.B. S.G.N. designed the project. Y.S., S.G.N. and V.L.R. interpreted the data and wrote the paper with input from G.W., J.B. and M.A. All authors contributed to discussion.

## Competing interests

The authors declare no competing interests.
