## [Peer Review File · Nature Communications]

Sources of dehydration fluids underneath the Kamchatka arcREVIEWER COMMENTS

Reviewer #1 (Remarks to the Author):

Shu et al. present a considerable TI isotope dataset for relatively primitive lavas from an across-arc transect in the Kamchatka arc. To constrain potential components in the mantle source of these arc lavas, they have also characterised a large set of new core samples from outbound of the arc that includes sediments and OIB-type basalts linked to the Hawaii-Emperor chain. The latter are also fully characterised for trace element and radiogenic isotope composition, which makes this a very valuable dataset. The authors note that the Kamchatka lavas have a heavier TI isotopic composition relative to MORB and local subducting sediments, which they explain by the release of TI from the OIB-type basalts in the subducting slab. They then proceed to investigate the pathways of slab-to-mantle wedge mass transfer as a function of slab depth using mainly trace element systematics.

I appreciate the large amount of high-quality new data and the careful planning of this project by including new drill core samples, as well as the effort to combine novel (TI isotopes) with traditional (trace elements, Pb isotopes) geochemical proxies. The manuscript is well written and though provoking. Nevertheless, it does leave me with mixed feelings. Mainly, the paper feels similar to recent previous papers by this group that also looked at TI isotopes in arcs (Shu et al., GCA 2017; Nielsen et al., JVGR 2017). I can't help but feel that the same results and models are reiterated in this manuscript, with the exception that the Kamchatka arc lavas trend towards slightly heavier TI isotope composition due to the involvement of the OIB-type basalts in the slab at this location.

Though I am in general happy to accept this interpretation (with a caveat listed in my main comments below), I am less convinced with the stepwise dehydration model as presented here. It is based near-exclusively on ambiguous trace element constraints and TI actually does not really inform on these reactions. Moreover, I have concerns about some of the assumptions in this model, as outlined in detail below.

Overall, I feel that the abundance of high-quality isotope data most definitely warrant publication in an appropriate geochemical journal, but that the current interpretation needs some rethinking and/or better argumentation. Also, I feel better justification why this study is novel with respect to previous papers by these authors might be beneficial.

Main comments

The role of phengite forms is key in the model; it is presumed to be the main host for TI and a residual phase at lower pressure (EVF and CKD) and breaks down to release TI at higher pressure (SR). There are several potential problems with this interpretation that require attention. Foremost, is there any independent evidence that phengite will be present in meta-HESC and meta-AOC at this location? A protolith with a considerable K₂O content is needed to stabilise phengite at sub-arc metamorphic

conditions. Experimental studies show phengite to be present in most metasediment compositions (i.e., high K₂O) but not in metabasalt, even at bulk K₂O contents up to 1.2 wt.% (Schmidt et al, EPSL 2004; Klemme et al., GCA 2002). As the authors convincingly argue that there is only a very thin to absent layer of sediments subducting at this sector of the arc, how realistic is it that phengite is present in either meta-oceanic crust or a melange? Where is the required K₂O coming from? Notably, experimental studies do not find phengite as a residual phase during melange melting studies (Codillo et al. Nat Comms 2017; Cruz-Uribe et al. Geology 2018).

Moreover, I find it difficult to bring the high Tl contents of the EVF and CKD lavas in agreement with phengite being the main host of Tl. If phengite retains Tl, then why is there not a clear difference in Tl content between EVF/CKD and the SR centres that are apparently influenced by phengite breakdown? Also, if a significant fraction of Tl is held back by phengite, there is the possibility of Tl isotope fractionation between phengite and a melt/fluid. For very incompatible behaviour the mass balance is in favour of the fluid/melt, meaning that it is reasonable to assume that the Tl isotope composition of the source equals that of the partial melt or fluid, but if Tl is compatible in a phase, fractionation factors need to be taken into account and the partial melt or fluid may very well be fractionated relative to the source.

I find the trace element evidence for the various mineral breakdown reactions to be a bit overstated as in reality most samples overlap for the trace element ratios shown in Figure 7.

The Pb isotope offset between the xenoliths and lavas (Fig. 5) is surprising. If the xenoliths represent the metasomatised mantle wedge, arc lavas would be expected to overlap with the xenoliths. Could the shift to higher ²⁰⁸/₂₀₄Pb be attributed to crustal assimilation? If so, could this affect Tl isotope compositions? Or is there an alternative explanation for this offset?

I am not fully convinced that having HESC basalts in the source is the only or simplest explanation for the Tl isotope data. The authors convincingly argue for an Indian MORB-like mantle wedge composition based on Pb isotopes. Are there any Tl isotope data for Indian MORB? Is it conceivable that Indian MORB might be isotopically heavy through sediments in its source, as is suggested by radiogenic isotope data? Might a mantle wedge with a heavy Tl isotope composition not be an equally parsimonious explanation for the Kamchatka data?

Throughout the manuscript, “ratios” is often superfluous, e.g. in “Ce/Tl ratios”, “Nb/La ratios”. The / already signifies a ratio, so adding “ratio” is not needed. Better would simply be e.g., “higher Ce/Tl”, “lower Nb/La” etc.

The authors are encouraged to adhere to metrological guidelines in reporting isotope data and uncertainties. For examples, use “reference material” instead of “standard”, 2s for two times the sample

standard deviation (not sd), correct reporting of uncertainties (long-term reproducibility is not a correct term as it is not defined).

Line-specific comments

Line 36. “stable” is superfluous here. By the way, neither isotope of Tl reported here is technically stable; they are both observationally stable, which is not the same.

160. See comment on line 36.

161. Reference material, not standard

162. Why not use delta notation, which is the conventional and recommended way (by IUPAC) to report mass-dependent isotope variations?

165 and elsewhere. Be careful with significant figures for the Tl concentrations. In line 446, a 2s uncertainty on Tl concentrations of ca. 15% is listed, so using two significant figures seems more appropriate.

129-131. It would be useful to mention the thickness of the HESC basalts here. How much HESC basalt is there relative to normal MORB?

173-175. Though oddly they don't have higher Tl contents? Or is it the absence of “normal” Tl in sediments that makes the Mn oxide signature more prominent?

183-184. This is an odd proxy to use as it is not very sensitive given the much stronger relative enrichment of Tl in these Mn oxides compared to Mn.

185-188. I was surprised why the Tl isotope data for the HESC samples were not directly compared to Hawaiian samples in the text and figures, as data for Hawaiian basalts are clearly available. Upon checking reference 57 it became clear that the HESC samples are notably heavier than the Hawaiian samples. Hence, this statement is a bit coy, and unnecessary. Why not allow for some Mn oxide

deposition in these basalts post-emplacement? It won't change the story at all. The direct data for the HESC samples are compelling without them being identical to Hawaiian samples.

199. This is not the full range.

207. Thallium (don't start a sentence with an element symbol)

210. Here I am also surprised by the lack of comparison to these other arc data in the text and figures.

210. Which mineral(s) incorporate and fractionate Tl during crystallisation to cause this inflection? I would say it is highly incompatible and hence should not fractionate.

212. These ratios are so source-sensitive and hence variable among arc basalts that this argument is rather weak.

226. Churikova et al. (2001) instead of "a previous study".

248. "Pacific upper mantle" is confusing as it is the Pacific plate that subducts. What I think the authors mean is a mantle wedge with a Pb isotope composition akin to Pacific MORB.

251. Similar to the previous comment; a mantle wedge with Pb isotope composition similar to Indian MORB.

263-265. Though these are only literature data; none of the new data have $^{206}/^{204}\text{Pb} < 18.2$.

268. Mass balance – is that bulk, or melts/fluids? This is mentioned in line 275 but stating that already in this sentence might be better.

278-279. Indeed. But what is possible, is calculating the mass balance of Pb, i.e. how much Pb is derived from each component. That alleviates the requirement of knowing absolute Pb concentrations of the components.

288. “almost entirely similar” is very vague; the samples extend to notably heavier Tl isotope compositions. Why not clarify by stating that the samples range from typical MORB values to heavier?

290. How is this sub-arc mantle wedge composition constrained? I guess it is average MORB, but how appropriate is that? See major comment on a possible alternative explanation by adopting an isotopically heavier mantle wedge composition.

291-292. If sediments and the mantle wedge have identical Tl isotope compositions, Tl isotopes cannot be used as an argument that the sediment contribution is small. I don't disagree with that interpretation, but only based on the Pb isotope data. Thallium in this case does not tell you anything about a sediment contribution.

328-329. Why would fluids in equilibrium with residual phengite be Tl-poor, but partial melts with residual rutile be Tl-rich? If “phengite is the primary mineral that can accommodate Tl” (line 321), it will retain the majority of Tl to greater depth. Then where is all the Tl in the EVF coming from?

339-340. The trace element evidence is a bit overstated as in reality most samples overlap for the trace elements shown in Figure 7.

394. have been identified

429, 433. Please use element symbols unless when it is the first word of a sentence.

Figure 3. Perhaps clearly show the samples excluded from the rest of the discussion here, for instance through filled v open symbols? In general, how important is showing the individual volcanic centres? Perhaps consider using a single symbol and colour for each of the three sample groups (instead of volcanic centre) to make the figures less cluttered?

Reviewer #2 (Remarks to the Author):

Review of: "Sources of dehydration fluids underneath the Kamchatka arc"

This manuscript showcases the application of an increasingly explored novel stable isotope system to reveal the dehydration and fluid mediation history of magma sources in the well-characterised Kamchatka arc subduction zone. The study thus combines new thallium isotope data with published conventional stable and radiogenic isotope and trace element data to discuss if fluids were derived from either altered oceanic crust, the mantle wedge, or *mélange* rocks, i.e. a mechanical mixture of subduction zone components. In brief, I think that this is an excellent work and the results have a high potential to help our understanding of processes operating in subduction zones, one of the most important locations of mass transfer on Earth, and hence, it is well suited for publication in *Nature Communications*. My brief review below should not be mistaken for a lack of interest in the work or a lack of attention while reading it – the manuscript is very well structured and written, the data are of high quality and seem reliable, and I can't fault the interpretations, thus I have relatively little to comment. The study is provocative and surely will spark discussions among the subduction zone research community.

Line 128: selected for this study

Line 230 (and supplementary): Why was the cut-off specifically chosen at $Dy/Yb > 2$? If the REE shape in lavas from this volcano is controlled by garnet in the source, I would expect the whole $Dy/Yb = 1.8-2.4$ range to show this. Meaning, the low Ce/Tl samples from Ichinsky could equally have formed in the presence of garnet (the Dy/Yb values just below 2.0 don't speak against it). It is just not a very convincing argument to divide the samples based on $Dy/Yb = 1.8-2.0$ compared to $Dy/Yb > 2$ samples as both can be regarded as garnet signature.

Trace element ratios in CKD lavas (line 339-340) and SR lavas (lines 369-370):

I don't think the comparison of trace element ratios between CKD and EVF lavas fits the observation well in some cases. For example it is stated that Ba/Th, Cs/U or Cs/Pb would be lower. Judging by Figure 6 and 7 the values for both volcanic suites appear quite similar. U/K ratios are in the same range, too.

In the case of SR volcanoes: not all volcanoes have high Ba/Th (Ichinsky doesn't) and Pb/Tl generally is in the same range as EVF and CKD.

I do understand that the word count must be kept down in the main text and there is no space to give a detailed comparison of all trace element ratios. But maybe just highlight those where the differences are actually obvious. This won't diminish the interpretations.

Line 542: This sentence reads a bit strangely because it has “beneath SR” and “underlying CKD” in it. Please rephrase to clarify that the mélange was removed before fluids contributing to SR are released.

Figures:

Figure 2: Here and in the following figures you use a very similar colour for the HESC basalt and MORB. I am aware that both are basalts, but I would suggest using a stronger contrast in colour.

Figure 4: Is there any reason why you use the TI isotope composition of the upper mantle here compared to MORB in all of the other figures?

And: You have ordered the volcanoes from North to South. This, however, visually connotes that the SR volcanoes are geographically closest to Site 192 sediments. I would suggest grouping them according to slab depths with the deepest at the left and the shallowest to the right towards the sediment site. (This is really nitpicking and just a suggestion!).

Good luck with the revision.

Mathias Schannor

Berlin, January 2022

Reviewer #3 (Remarks to the Author):

The study “Sources of dehydration fluids underneath the Kamchatka arc” uses TI isotopes to investigate the fluid sources of arc magmas in Kamchatka. I enjoyed reading the manuscript, it is an important contribution that gives new insights into the fluids in a cold subduction zone. The only thing that I am wondering is the K budget. The authors show repeatedly that sediments don’t play an important role in the arc magma generation but also show that several rocks belong to the high-K series and that phengite plays an integral role in the TI release. How can there be significant amounts of phengite without a reasonable K source (sediments)? Oceanic basalts are generally very low in K (<0.5 wt%) and low degree mantle melting still produces more Na₂O than K₂O (10 times more), so there needs to be K-enrichment.

Comments:

Lines 79-80: How do you generate high-K arc basalts without significant sediment input? Sediments are a vital K source and neither the oceanic plate nor the asthenospheric mantle has high K contents.

Line 312: If the sediment component is minimal in the arc magma source does this necessarily mean that sediments cannot contribute fluids to the forearc?

Lines 321-322: How much phengite forms from K-poor (tholeiite)- basaltic crust? Phengite contains ~10 wt% K₂O and basalts have <0.5 wt%. At high PT, CPx also incorporates K, lowering the budget available for phengite. Would <1% phengite be enough to accommodate the TI?

The original reviewer's comments are copied in black, and our responses are in red. Line numbers in red refer to line numbers in the revised "Tracked changes" manuscript.

REVIEWER COMMENTS

Reviewer #1 (Remarks to the Author):

Shu et al. present a considerable Tl isotope dataset for relatively primitive lavas from an across-arc transect in the Kamchatka arc. To constrain potential components in the mantle source of these arc lavas, they have also characterised a large set of new core samples from outboard of the arc that includes sediments and OIB-type basalts linked to the Hawaii-Emperor chain. The latter are also fully characterised for trace element and radiogenic isotope composition, which makes this a very valuable dataset. The authors note that the Kamchatka lavas have a heavier Tl isotopic composition relative to MORB and local subducting sediments, which they explain by the release of Tl from the OIB-type basalts in the subducting slab. They then proceed to investigate the pathways of slab-to-mantle wedge mass transfer as a function of slab depth using mainly trace element systematics.

I appreciate the large amount of high-quality new data and the careful planning of this project by including new drill core samples, as well as the effort to combine novel (Tl isotopes) with traditional (trace elements, Pb isotopes) geochemical proxies. The manuscript is well written and though provoking. Nevertheless, it does leave me with mixed feelings. Mainly, the paper feels similar to recent previous papers by this group that also looked at Tl isotopes in arcs (Shu et al., GCA 2017; Nielsen et al., JVGR 2017). I can't help but feel that the same results and models are reiterated in this manuscript, with the exception that the Kamchatka arc lavas trend towards slightly heavier Tl isotope composition due to the involvement of the OIB-type basalts in the slab at this location.

Though I am in general happy to accept this interpretation (with a caveat listed in my main comments below), I am less convinced with the stepwise dehydration model as presented here. It is based near-exclusively on ambiguous trace element constraints and Tl actually does not really inform on these reactions. Moreover, I have concerns about some of the assumptions in this model, as outlined in detail below.

Overall, I feel that the abundance of high-quality isotope data most definitely warrant publication in an appropriate geochemical journal, but that the current interpretation needs some rethinking and/or better argumentation. Also, I feel better justification why this study is novel with respect to previous papers by these authors might be beneficial.

We appreciate the constructive comments provided by the reviewer, and we address concerns related to stepwise dehydration in our detailed response. Here, we take the opportunity to clarify the novelty of our study, in particular compared to the previous studies mentioned by the reviewer

- 1) **We identify the sources of fluids coming out of an oceanic crust uncontaminated by sediment input.** Such a tectonic setting is rare, and the Kamchatka arc study has been designed to provide novel constraints on fluids solely coming out of the oceanic crust, without the additional complication of large sediment input. In previous studies based on the Ryukyu arc-Okinawa Trough system, Aleutian, Tonga-Kermadec, and Central American arcs (Nielsen et al., 2017; Shu et al., 2017; Nielsen et al., 2016), we primarily discussed the two contrasting processes of sediment addition to the mantle wedge (as bulk sediments or sediment melts).
- 2) **As a fluid-mobile element, we demonstrate that Tl can directly track the source of fluids that infiltrate the Kamchatka sub-arc mantle wedge.** We disagree that Tl does not inform on dehydration reactions. Both Tl isotopes and Tl concentrations record the progressive removal of the HESC and altered oceanic crust, effectively tracking the changing composition of the source of fluids as the slab subducts to deeper depths. As shown in Fig. 7, from EVF ($\epsilon^{205}\text{Tl} = -4.7$ to $+3.6$), through CKD ($\epsilon^{205}\text{Tl} = -1.1$ to $+1.1$), to SR ($\epsilon^{205}\text{Tl} = -2.0$ to -0.2), Tl isotope characteristics reveal stepwise removal of HESC and altered oceanic crust ($\epsilon^{205}\text{Tl} = -4$ to $+4$) from EVF to CKD, and HESC components ($\epsilon^{205}\text{Tl} > 0$) from CKD to SR (Lines 429-441). Sediments usually control Tl isotope variations in previously investigated arcs (Nielsen et al., 2017; Shu et al., 2017; Nielsen et al., 2016), but the Kamchatka window offers the opportunity to study fluids and fluid-mobile elements coming primarily from the oceanic crust.
- 3) **To our knowledge, no such integrated study exists in the literature.** The study provides an integrated view of Tl isotope variations, combined with an extensive array of

geochemical tracers (elemental and isotopic), on both the inputs and outputs of a subduction zone dominated by oceanic crust recycling.

We have re-emphasized the novel aspects of our study and how it differs from our previous studies in the revised manuscript (Lines 92-96).

Main comments

The role of phengite forms is key in the model; it is presumed to be the main host for Tl and a residual phase at lower pressure (EVF and CKD) and breaks down to release Tl at higher pressure (SR). There are several potential problems with this interpretation that require attention. Foremost, is there any independent evidence that phengite will be present in meta-HESC and meta-AOC at this location? A protolith with a considerable K₂O content is needed to stabilise phengite at sub-arc metamorphic conditions. Experimental studies show phengite to be present in most metasediment compositions (i.e, high K₂O) but not in metabasalt, even at bulk K₂O contents up to 1.2 wt.% (Schmidt et al, EPSL 2004; Klemme et al., GCA 2002). As the authors convincingly argue that there is only a very thin to absent layer of sediments subducting at this sector of the arc, how realistic is it that phengite is present in either meta-oceanic crust or a melange? Where is the required K₂O coming from?

The reviewer is concerned about the presence of phengite in meta-basalts. We agree that fresh MORB contains low K₂O. However, analyses of the HESC basalts (Table S4) reveal these to have similar K₂O concentrations to the subducted sediments (mostly between 1 and 2 wt %), which is much higher than fresh MORB. The high K concentrations in the HESC basalts are likely partially due to their origin from a plume source and partially as a result of hydrothermal alteration at low temperature, which is known to cause significant K enrichment (Parendo et al., 2017; Kelley et al., 2003; Staudigel et al., 1995). It is, therefore, not particularly surprising that the subducted HESC would form phengite at appropriate slab pressure and temperatures. Furthermore, the presence of phengite and its likely control of Tl abundances in exhumed MORB-like eclogites (analogs of altered oceanic crust subducted underneath the HESC) has also been observed in several studies on eclogites from several localities globally (Shu et al., 2022; Urann et al., 2020; Shu et al., 2019). We have now added these arguments to the revised manuscript (Lines 337-343 and 419-422).

Notably, experimental studies do not find phengite as a residual phase during mélangé melting studies (Codillo et al. Nat Comms 2017; Cruz-Uribe et al. Geology 2018).

The reviewer is right that phengite has not been reported as a residual phase in these two papers. However, we note that starting materials in the Codillo et al. (2018) study is primarily peridotite with 5–15 % mélangé rock, so that phengite would not necessarily be expected to stabilize. We also emphasize that the starting mélangé used in Cruz-Uribe et al. (2018) is a chlorite-omphacite rock that in fact does contain phengite (along with epidote, rutile, titanite, apatite, and tourmaline). The high degrees of melting in Cruz-Uribe et al. (2018) likely precluded preservation of the phengite in the experimental residue. In addition, phengite is a common phase in jadeites and has been reported in exhumed mélangé zones (e.g., Marschall and Schumacher, 2012; Sorensen et al., 1997). Thus, the presence of phengite in meta-oceanic crust and mélanges is not uncommon. This point has been clarified in the revised manuscript (Lines 337-343 and 391-392).

Moreover, I find it difficult to bring the high Tl contents of the EVF and CKD lavas in agreement with phengite being the main host of Tl. If phengite retains Tl, then why is there not a clear difference in Tl content between EVF/CKD and the SR centres that are apparently influenced by phengite breakdown? Also, if a significant fraction of Tl is held back by phengite, there is the possibility of Tl isotope fractionation between phengite and a melt/fluid. For very incompatible behaviour the mass balance is in favour of the fluid/melt, meaning that it is reasonable to assume that the Tl isotope composition of the source equals that of the partial melt or fluid, but if Tl is compatible in a phase, fractionation factors need to be taken into account and the partial melt or fluid may very well be fractionated relative to the source.

We want to clarify this point.

We agree with the reviewer that the differences in Tl concentration between the three regions are probably smaller than what one might expect if phengite in a homogenous source controlled all three regions (SR is on average ~33% higher than EVF and CKD, see Fig. 3). However, it should be pointed out that the source material for EVF/CKD and SR regions are different because the SR region likely sources a component dominated by lower oceanic crust whereas EVF/CKD is dominated by Tl-rich HESC material. The Tl concentration of lower oceanic crust is likely lower

than HESC, which complicates the direct use of Tl concentrations in the way suggested by the reviewer. In order to avoid these uncertainties we instead use the relative partitioning of Cs and Tl in phengite, which, as explained in the manuscript, translates into the observed differences in Cs/Tl between EVF/CKD and SR (see Fig. 7b).

We added some new text outlining the above arguments (Lines 358-363 and 410-411).

The reviewer also mentions the possibility of Tl isotope fractionation between phengite and fluids/melts. However, our previous studies have shown that Tl concentration and isotope data for samples of subducted oceanic crust are inconsistent with any systematic Tl isotope fractionation during partial removal of Tl from a phengite-bearing assemblage via either fluids or melts (Shu et al., 2022; Shu et al., 2019). We have added this argument in the revised manuscript (Lines 343-346).

I find the trace element evidence for the various mineral breakdown reactions to be a bit overstated as in reality most samples overlap for the trace element ratios shown in Figure 7. We thank the reviewer for this point, which was also raised by Reviewer 2. As suggested by reviewer 2, we now only discuss the elemental ratios that show the clearest differences between the three regions (i.e., Pb/Tl, Cs/Tl, Sr/Nd). We have revised Fig. 7 accordingly using a single symbol and color for each of the three sample groups (instead of volcanic centers) to make the figure less cluttered and added the average values of EVF, CKD and SR samples. We have also added and rephrased sentences to re-emphasize how each region displays a unique combination of Tl isotope and trace element variations (Lines 362-363 and 372-373).

The Pb isotope offset between the xenoliths and lavas (Fig. 5) is surprising. If the xenoliths represent the metasomatised mantle wedge, arc lavas would be expected to overlap with the xenoliths. Could the shift to higher $^{208}\text{Pb}/^{204}\text{Pb}$ be attributed to crustal assimilation? If so, could this affect Tl isotope compositions? Or is there an alternative explanation for this offset? First, the offset is actually only in $^{206}\text{Pb}/^{204}\text{Pb}$, but arc lavas and xenoliths overlap in $^{208}\text{Pb}/^{204}\text{Pb}$. Crustal assimilation would typically result in arc lavas showing an increase in both $^{208}\text{Pb}/^{204}\text{Pb}$ and $^{206}\text{Pb}/^{204}\text{Pb}$ ratios, not in a decrease of $^{206}\text{Pb}/^{204}\text{Pb}$ and constant $^{208}\text{Pb}/^{204}\text{Pb}$ (e.g. Thirlwall et al., 1996). Second, the reviewer is correct that the mismatch between Pb isotope composition in

Kamchatka mantle xenoliths and arc lavas would not be expected if these xenoliths represented a truly unmodified mantle wedge. It should be noted that the 11 xenoliths which all come from Saha et al. (2005) exhibit extremely high Pb and Ba concentrations relative to normal mantle values (>10 times higher), which suggests that these do not represent the unmodified mantle wedge, but one that was modified by Pb-rich fluids. These xenoliths have far higher Pb contents than the refractory mantle xenoliths from Avachinsky volcano, for which there is sadly no Pb isotope data (Ionov, 2010). In addition, Sr and Nd isotope data on leached clinopyroxene from the mantle xenoliths reveal some peridotites are affected by Sr-rich and Nd-poor fluids (Churikova et al., 2001). Therefore, the xenoliths are pervasively affected by fluids. We have clarified these points in the text (Lines 267-272).

I am not fully convinced that having HESC basalts in the source is the only or simplest explanation for the Tl isotope data. The authors convincingly argue for an Indian MORB-like mantle wedge composition based on Pb isotopes. Are there any Tl isotope data for Indian MORB? Is it conceivable that Indian MORB might be isotopically heavy through sediments in its source, as is suggested by radiogenic isotope data? Might a mantle wedge with a heavy Tl isotope composition not be an equally parsimonious explanation for the Kamchatka data?

There is no direct Tl isotope data available for Indian MORB. However, extensive Tl isotope data sets have been published for the Ryukyu and Mariana arcs (Shu et al., 2017; Prytulak et al., 2013), in which it is generally accepted that Indian MORB dominates the mantle wedge (as we also outline in the original manuscript). In these arcs, the Indian MORB mantle was found to exhibit $\epsilon^{205}\text{Tl} \sim -2.0$, and we see no reason to assume the Indian MORB mantle would be different underneath the Kamchatka arc. In addition, if the Indian MORB mantle was locally, coincidentally heavy in the Kamchatka region only, it should be heavy throughout EVF, CKD and SR, which is not the case. We have added the argument in the revised manuscript (Lines 305-308).

Throughout the manuscript, “ratios” is often superfluous, e.g. in “Ce/Tl ratios”, “Nb/La ratios”. The / already signifies a ratio, so adding “ratio” is not needed. Better would simply be e.g., “higher Ce/Tl”, “lower Nb/La” etc.

Corrected.

The authors are encouraged to adhere to metrological guidelines in reporting isotope data and uncertainties. For examples, use “reference material” instead of “standard”, 2s for two times the sample standard deviation (not sd), correct reporting of uncertainties (long-term reproducibility is not a correct term as it is not defined).

Corrected.

Line-specific comments

Line 36. “stable” is superfluous here. By the way, neither isotope of Tl reported here is technically stable; they are both observationally stable, which is not the same.

160. See comment on line 36.

Corrected.

161. Reference material, not standard

Corrected.

162. Why not use delta notation, which is the conventional and recommended way (by IUPAC) to report mass-dependent isotope variations?

Tl isotope compositions usually are reported in epsilon notation because Tl isotopes were originally developed as a cosmochemical extinct radiogenic isotope system (Pb-205 decays to Tl-205 with ~15Myr half-life), where the convention has been to use epsilon units. Hence, in order to facilitate comparisons between all published data sets, we have decided to stick with the epsilon notation.

165 and elsewhere. Be careful with significant figures for the Tl concentrations. In line 446, a 2s uncertainty on Tl concentrations of ca. 15% is listed, so using two significant figures seems more appropriate.

Corrected. We use two significant figures for the Tl concentrations in the revised manuscript.

129-131. It would be useful to mention the thickness of the HESC basalts here. How much

HESC basalt is there relative to normal MORB?

There is, to our knowledge, no direct information about the relative thicknesses of HESC and underlying AOC. The seafloor in front of the Kamchatka arc is ~5.5km below sea level, whereas the Meiji seamount is ~3km below sea level. The ~1km sediment cover on the Meiji seamount would suggest a thickness of at least 1.5km for the HESC material. However, due to lithospheric loading, the HESC thickness is likely significantly larger than 1.5km. We have included the description of the thickness of HESC basalts in the revised manuscript (Lines 135-138).

173-175. Though oddly they don't have higher Tl contents? Or is it the absence of "normal" Tl in sediments that makes the Mn oxide signature more prominent?

This is a good observation. Most likely the low concentrations are at least partially due to the relatively high Ca carbonate contents of these sediments (as seen by high Ca and Sr concentrations in Table S4), which contains essentially no Tl and, hence, dilutes the Tl concentration. We have added this additional information to the main text (Lines 180-181) and added a reference for the low Tl contents of carbonate-rich sediments.

183-184. This is an odd proxy to use as it is not very sensitive given the much stronger relative enrichment of Tl in these Mn oxides compared to Mn.

Good suggestion. We have revised Fig. S1 by deleting the Fe/Mn proxy as we agree this might not be likely to register such small amounts of Mn oxide material.

185-188. I was surprised why the Tl isotope data for the HESC samples were not directly compared to Hawaiian samples in the text and figures, as data for Hawaiian basalts are clearly available. Upon checking reference 57 it became clear that the HESC samples are notably heavier than the Hawaiian samples. Hence, this statement is a bit coy, and unnecessary. Why not allow for some Mn oxide deposition in these basalts post-emplacement? It won't change the story at all. The direct data for the HESC samples are compelling without them being identical to Hawaiian samples.

We agree that a direct comparison between HESC and Hawaii data is warranted, which we have added to Figure S1. We have also added information regarding the possible origin of the difference in Tl isotope compositions between HESC and Hawaiian lavas (Lines 192-197).

199. This is not the full range.

These values refer to the weighted average Tl isotope compositions of Site 192 and Hole 881C sediments. We have rephrased this sentence for clarity (Line 208).

207. Thallium (don't start a sentence with an element symbol)

Corrected.

210. Here I am also surprised by the lack of comparison to these other arc data in the text and figures.

In Fig. S5, we compared the Ce/Tl for the Kamchatka arc and other arcs analyzed for Tl concentrations. We have also added a comparison of Tl isotope values (Lines 205-209).

210. Which mineral(s) incorporate and fractionate Tl during crystallisation to cause this inflection? I would say it is highly incompatible and hence should not fractionate.

We agree with the reviewer that Tl is highly incompatible and hence should not fractionate during the early stage of magma evolution. However, given that Tl often behaves similarly to alkali metals K, Rb and Cs, Tl may potentially be fractionated by mica or feldspar at late stages of magma differentiation. Therefore, the samples analyzed in this study exclude the highly fractionated samples with low MgO and high SiO₂ contents (Lines 220-221).

212. These ratios are so source-sensitive and hence variable among arc basalts that this argument is rather weak.

It is still useful to compare the general range of Ce/Tl and Th/Rb values in arc lavas to those measured for a given sample, to potentially identify severely altered samples. Previous studies have shown that coupled $Ce/Tl > 2000$ and $Th/Rb > 0.2$ for both OIBs and arc lavas are strongly suggestive of subaerial alteration (Nielsen et al., 2016; Nielsen et al., 2006). It is correct that minor effects from alteration would not be detected since the source variations of the two ratios are relatively large. In the present study, all Kamchatka samples display $Ce/Tl < 1100$ and $Th/Rb < 0.1$ (Fig. S5), which implies minor to no subaerial alteration. We have made a slight change in the wording in the supplement where it is already stated that samples were collected specifically

to be as fresh as possible and, hence, are unlikely to be affected by subaerial alteration (Lines 65-74 in Supplementary).

226. Churikova et al. (2001) instead of “a previous study”.

Corrected.

248. “Pacific upper mantle” is confusing as it is the Pacific plate that subducts. What I think the authors mean is a mantle wedge with a Pb isotope composition akin to Pacific MORB.

Corrected.

251. Similar to the previous comment; a mantle wedge with Pb isotope composition similar to Indian MORB.

Corrected.

263-265. Though these are only literature data; none of the new data have $^{206}\text{Pb}/^{204}\text{Pb} < 18.2$.

The reviewer is right. Even though the new data have $^{206}\text{Pb}/^{204}\text{Pb} > 18.2$, the mixing model between Pacific MORB mantle and HESC basalts or sediments cannot account for either our new or the older literature Pb isotope data (Fig. 5). We have clarified in the text that low $^{206}\text{Pb}/^{204}\text{Pb}$ are found in literature Pb isotope data (Lines 272-274).

268. Mass balance – is that bulk, or melts/fluids? This is mentioned in line 275 but stating that already in this sentence might be better.

Agreed. We have rephrased this sentence (Lines 277-280).

278-279. Indeed. But what is possible, is calculating the mass balance of Pb, i.e. how much Pb is derived from each component. That alleviates the requirement of knowing absolute Pb concentrations of the components.

The reviewer is correct and this is an exercise that has previously been performed. Based on the assumption that the fluid does not carry any significant amount of HFSE and HREE, the fluid-absent source can then be estimated by fitting a line through the HFSE and HREE concentrations (Pearce et al., 1995). According to this approach, Churikova et al. (2001) shows that more than

90% of Pb in the Kamchatka samples is derived from the slab fluid component (AOC/HESC), and the sediment component is less than 10%. We have added this discussion in the main text (Lines 286-288).

288. “almost entirely similar” is very vague; the samples extend to notably heavier Tl isotope compositions. Why not clarify by stating that the samples range from typical MORB values to heavier?

Agreed. We have rephrased this sentence (Lines 297-299).

290. How is this sub-arc mantle wedge composition constrained? I guess it is average MORB, but how appropriate is that? See major comment on a possible alternative explanation by adopting an isotopically heavier mantle wedge composition.

Please see our response to the major comment above. We have also added a sentence to characterize the Tl isotope composition of the Indian MORB mantle (Lines 305-308).

291-292. If sediments and the mantle wedge have identical Tl isotope compositions, Tl isotopes cannot be used as an argument that the sediment contribution is small. I don't disagree with that interpretation, but only based on the Pb isotope data. Thallium in this case does not tell you anything about a sediment contribution.

Sediments carry much higher Tl concentrations than HESC basalts (Fig. 2). So, if sediment is a significant subducted component underneath Kamchatka, the Tl isotope characteristics of HESC basalts would essentially be buffered by sediments. In this case, our Kamchatka samples would display a narrow Tl isotope range of $\epsilon^{205}\text{Tl}$ values around -2, which is not observed. Therefore, our Tl isotope compositions, and Pb isotope data, both support a sediment-poor source of Kamchatka arc lavas. We have rephrased the text to clarify the argument (Lines 299-305).

328-329. Why would fluids in equilibrium with residual phengite be Tl-poor, but partial melts with residual rutile be Tl-rich? If “phengite is the primary mineral that can accommodate Tl” (line 321), it will retain the majority of Tl to greater depth. Then where is all the Tl in the EVF coming from?

We agree with the reviewer that this on the surface seems counterintuitive but we refer to a

detailed answer to the comment on phengite that addresses this concern. In addition, we did not mention “partial melts with residual rutile”. We suppose the reviewer is talking about “partial melts with residual phengite”. We also added sentences for clarity (Lines 354-358).

339-340. The trace element evidence is a bit overstated as in reality most samples overlap for the trace elements shown in Figure 7.

Please see our response to the major comment above.

394. have been identified

Corrected.

429, 433. Please use element symbols unless when it is the first word of a sentence.

Corrected.

Figure 3. Perhaps clearly show the samples excluded from the rest of the discussion here, for instance through filled v open symbols? In general, how important is showing the individual volcanic centres? Perhaps consider using a single symbol and colour for each of the three sample groups (instead of volcanic centre) to make the figures less cluttered?

This is a good idea. We revised Fig. 3 and all subsequent figures.

Reviewer #2 (Remarks to the Author):

Review of: “Sources of dehydration fluids underneath the Kamchatka arc”

This manuscript showcases the application of an increasingly explored novel stable isotope system to reveal the dehydration and fluid mediation history of magma sources in the well-characterised Kamchatka arc subduction zone. The study thus combines new thallium isotope data with published conventional stable and radiogenic isotope and trace element data to discuss if fluids were derived from either altered oceanic crust, the mantle wedge, or mélange rocks, i.e. a mechanical mixture of subduction zone components. In brief, I think that this is an excellent work and the results have a high potential to help our understanding of processes operating in

subduction zones, one of the most important locations of mass transfer on Earth, and hence, it is well suited for publication in Nature Communications. My brief review below should not be mistaken for a lack of interest in the work or a lack of attention while reading it – the manuscript is very well structured and written, the data are of high quality and seem reliable, and I can't fault the interpretations, thus I have relatively little to comment. The study is provocative and surely will spark discussions among the subduction zone research community.

We thank Review 2 for their positive comments.

Line 128: selected for this study

Corrected.

Line 230 (and supplementary): Why was the cut-off specifically chosen at $Dy/Yb > 2$? If the REE shape in lavas from this volcano is controlled by garnet in the source, I would expect the whole $Dy/Yb = 1.8-2.4$ range to show this. Meaning, the low Ce/Tl samples from Ichinsky could equally have formed in the presence of garnet (the Dy/Yb values just below 2.0 don't speak against it). It is just not a very convincing argument to divide the samples based on $Dy/Yb = 1.8-2.0$ compared to $Dy/Yb > 2$ samples as both can be regarded as garnet signature.

This is a fair point. In theory, all Ichinsky samples may be variably affected by a garnet-rich, OIB-like component. However, on average, Ichinsky samples that display $Dy/Yb > 2$ also show elevated Ce/Tl (on average 1331) compared to samples that display $Dy/Yb < 2$ (on average 334). Elevated Ce/Tl has been shown to reflect the presence of a garnet-rich, subducted crust component in OIB (Blusztajn et al., 2018; Nielsen et al., 2007; Nielsen et al., 2006). Therefore, Ichinsky samples that most clearly track this OIB component, also suggested by previous studies (Churikova et al., 2001), are excluded from the dataset. The Ichinsky data that show lower Dy/Yb (in the 1.8–1.9 range) and low Ce/Tl are considered adequate for the purpose of this study. We have rephrased these sentences in the main text (Lines 238-243) and Supplementary (Lines 48-54) to clarify our interpretations.

Trace element ratios in CKD lavas (line 339-340) and SR lavas (lines 369-370):

I don't think the comparison of trace element ratios between CKD and EVF lavas fits the

observation well in some cases. For example it is stated that Ba/Th, Cs/U or Cs/Pb would be lower. Judging by Figure 6 and 7 the values for both volcanic suites appear quite similar. U/K ratios are in the same range, too.

In the case of SR volcanoes: not all volcanoes have high Ba/Th (Ichinsky doesn't) and Pb/Tl generally is in the same range as EVF and CKD.

I do understand that the word count must be kept down in the main text and there is no space to give a detailed comparison of all trace element ratios. But maybe just highlight those where the differences are actually obvious. This won't diminish the interpretations.

This is a very valid point, which was also commented on by Reviewer 1.

We agree that some of the groups only show weak differences and have removed the U/K and Ba/Th plots as these ratios showed the least systematic changes in the different groups. We have also revised the Figs. 6 and 7 using a single symbol and color for each of the three sample groups (instead of volcanic centers) to make the figures less cluttered. We have also added the average values of EVF, CKD and SR samples in the revised Fig. 6 and 7 to make it easier to appreciate the average changes in each region.

Line 542: This sentence reads a bit strangely because it has "beneath SR" and "underlying CKD" in it. Please rephrase to clarify that the mélangé was removed before fluids contributing to SR are released.

Agreed. We have rephrased this sentence (Lines 574-576).

Figures:

Figure 2: Here and in the following figures you use a very similar colour for the HESC basalt and MORB. I am aware that both are basalts, but I would suggest using a stronger contrast in colour.

This is a good suggestion. We have revised this figure.

Figure 4: Is there any reason why you use the Tl isotope composition of the upper mantle here compared to MORB in all of the other figures?

And: You have ordered the volcanoes from North to South. This, however, visually connotes that the SR volcanoes are geographically closest to Site 192 sediments. I would suggest grouping

them according to slab depths with the deepest at the left and the shallowest to the right towards the sediment site. (This is really nitpicking and just a suggestion!).

Agreed. We have replaced “Upper mantle” by “MORB”.

This is a good suggestion. We have revised this figure.

Good luck with the revision.

Mathias Schannor

Berlin, January 2022

Reviewer #3 (Remarks to the Author):

The study “Sources of dehydration fluids underneath the Kamchatka arc” uses Tl isotopes to investigate the fluid sources of arc magmas in Kamchatka. I enjoyed reading the manuscript, it is an important contribution that gives new insights into the fluids in a cold subduction zone. The only thing that I am wondering is the K budget. The authors show repeatedly that sediments don’t play an important role in the arc magma generation but also show that several rocks belong to the high-K series and that phengite plays an integral role in the Tl release. How can there be significant amounts of phengite without a reasonable K source (sediments)? Oceanic basalts are generally very low in K (<0.5 wt%) and low degree mantle melting still produces more Na₂O than K₂O (10 times more), so there needs to be K-enrichment.

We thank Review 3 for this supportive assessment.

Regarding the K source in the subducted slab, we agree that this is an important point. It was also commented on by Reviewer 1, and as pointed out above, the HESC material contains far more K than does average basalts. We believe that clarifying this in the main text (Lines 337-343) addresses the concerns of reviewer 3. Please also see our detailed answer in the response to reviewer 1.

Comments:

Lines 79-80: How do you generate high-K arc basalts without significant sediment input?

Sediments are a vital K source and neither the oceanic plate nor the asthenospheric mantle has high K contents.

We have implemented additional arguments to clarify that these meta-basalts are capable of stabilizing phengite as they contain similar K abundances as marine sediments (Lines 337-343).

Line 312: If the sediment component is minimal in the arc magma source does this necessarily mean that sediments cannot contribute fluids to the forearc?

The reviewer is correct that some minor amount of fluids/melts from sediments could play a role. However, we argue throughout the text that such contributions must be very minor as otherwise the TI isotopes would converge towards the sediment value, which is not observed.

Lines 321-322: How much phengite forms from K-poor (tholeiite)- basaltic crust? Phengite contains ~10 wt% K₂O and basalts have <0.5 wt%. At high PT, CPx also incorporates K, lowering the budget available for phengite. Would <1% phengite be enough to accommodate the TI?

Phengite is actually commonly observed in meta-basalts, as reported in studies of fresh eclogites from the Raspas Complex, Ecuador (Urann et al., 2020; John et al., 2010), meta-gabbro and glaucophane schist samples from Syros island, Greece (Marschall et al., 2006), eclogite and blueschist samples from the Tian Shan, northwest China (van der Straaten et al., 2008), eclogite and blueschist samples from Franciscan Complex, California (Sorensen et al., 1997) and eclogites from Dabie terrane, China (Guo et al., 2015). Therefore, it is reasonable to assume that it could also be present in the subducted oceanic crust under the Kamchatka arc. Also, Shu et al. (2019), which we have cited throughout the manuscript, showed that TI abundances in eclogites are strongly correlated with K, further implying that phengite, even at minor modal abundances, plays a key role in the TI budget of eclogites.

REVIEWERS' COMMENTS

Reviewer #1 (Remarks to the Author):

This is a revised version of the manuscript by Shu et al. It is clear that the authors have taken on board the comments made by me and other reviewers on the original submission. Most of these points have now been addressed in more detail, and the arguments for the progressive dehydration model are better substantiated by arguments and citations of relevant literature. This manuscript is in a publishable state and I have no further comments.

Reviewer #2 (Remarks to the Author):

I have looked at the revised manuscript and find that the authors have addressed all suggestions and comments raised by the reviewers. There are two small points mentioned below that should be corrected, but overall I recommend publication of the manuscript.

1) In lines 348-350 it is argued that melting of basalts is unlikely at depths of 90-110km below EVF. Contrastingly, in line 351 and following melting of a *mélange* is presented as alternative. I do not oppose this view, it reads however as if a *mélange* melts at depth where basalt melting is unlikely. Do you rather mean that *mélange* diapirs rise and melt due to decompression? If they melt at depths >90km, maybe add a reference showing that this is physically feasible. Or alternatively maybe add a sentence to explain that rising diapirs melt at shallower depths (as shown in Figure 8).

2) Supplementary file, line 66: surely Rb instead of Rd

Mathias Schannor

Berlin, 27/04/2022

Reviewer #3 (Remarks to the Author):

My comments have all been sufficiently addressed. I am looking forward to seeing this manuscript published.

The original reviewer's comments are copied in black, and our responses are in red. Line numbers in red refer to line numbers in the revised "Tracked changes" manuscript.

Reviewer #1 (Remarks to the Author):

This is a revised version of the manuscript by Shu et al. It is clear that the authors have taken on board the comments made by me and other reviewers on the original submission. Most of these points have now been addressed in more detail, and the arguments for the progressive dehydration model are better substantiated by arguments and citations of relevant literature. This manuscript is in a publishable state and I have no further comments.

We thank the reviewer for this supportive feedback.

Reviewer #2 (Remarks to the Author):

I have looked at the revised manuscript and find that the authors have addressed all suggestions and comments raised by the reviewers. There are two small points mentioned below that should be corrected, but overall I recommend publication of the manuscript.

1) In lines 348-350 it is argued that melting of basalts is unlikely at depths of 90-110km below EVF. Contrastingly, in line 351 and following melting of a mélangé is presented as alternative. I do not oppose this view, it reads however as if a mélangé melts at depth where basalt melting is unlikely. Do you rather mean that mélangé diapirs rise and melt due to decompression? If they melt at depths >90km, maybe add a reference showing that this is physically feasible. Or alternatively maybe add a sentence to explain that rising diapirs melt at shallower depths (as shown in Figure 8).

We thank the reviewer for this comment.

We have both added a reference documenting lower melting temperatures for mélangé than eclogitized oceanic crust as well as a sentence suggesting that rising mélangé diapirs might melt at shallow depths (Line 348-350).

2) Supplementary file, line 66: surely Rb instead of Rd

Corrected. Line 67 in Supplementary file.

Mathias Schannor

Berlin, 27/04/2022

Reviewer #3 (Remarks to the Author):

My comments have all been sufficiently addressed. I am looking forward to seeing this manuscript published.

We thank the reviewer.